# Antagonizing cholecystokinin A receptor in the lung attenuates obesity-induced airway hyperresponsiveness

Ronald Allan M. Panganiban[1], Zhiping Yang [1], Maoyun Sun[1], Chan Young Park[1], David I. Kasahara[1], Niccole Schaible[2], Ramaswamy Krishnan[2], Alvin T. Kho [3], Elliot Israel[4], Marc B. Hershenson[5], Scott T. Weiss[6], Blanca E. Himes[7], Jeffrey J. Fredberg [1], Kelan G. Tantisira[8], Stephanie A. Shore[1] & Quan Lu [1] ✉

Obesity increases asthma prevalence and severity. However, the underlying mechanisms are poorly understood, and consequently, therapeutic options for asthma patients with obesity remain limited. Here we report that cholecystokinin—a metabolic hormone best known for its role in signaling satiation and fat metabolism—is increased in the lungs of obese mice and that pharmacological blockade of cholecystokinin A receptor signaling reduces obesity-associated airway hyperresponsiveness. Activation of cholecystokinin A receptor by the hormone induces contraction of airway smooth muscle cells. In vivo, cholecystokinin level is elevated in the lungs of both genetically and diet-induced obese mice. Importantly, intranasal administration of cholecystokinin A receptor antagonists (proglumide and devazepide) suppresses the airway hyperresponsiveness in the obese mice. Together, our results reveal an unexpected role for cholecystokinin in the lung and support the repurposing of cholecystokinin A receptor antagonists as a potential therapy for asthma patients with obesity.

Asthma is a major chronic lung disease that affects over 300 million people worldwide[1]. A key feature of asthma is airflow limitation, often caused by hyperactive contraction of airway smooth muscle (ASM) cells[2]. Consequently, a major therapeutic goal for asthma is to reduce ASM constriction[2]. β-agonists—a mainstay asthma therapy—work by relaxing ASM to relieve bronchospasm[3,4]. However, long-term use of β-agonists, especially long-acting β-agonists (LABAs), can lead to paradoxically adverse effects, including loss of asthma control and even death[5,6]. Furthermore, standard controller therapy that uses inhaled corticosteroid in combination with a

β-agonist does not work well for certain patient populations, notably patients with obesity.

Obesity is a well-known risk factor for asthma: obesity increases both the prevalence and incidence of asthma and worsens asthma control[7]. It is estimated that in the United States alone, obesity leads to approximately 250,000 new cases of asthma each year[8]. In general, asthma patients with obesity have more symptoms, use more healthcare, and have worse quality of life[7]. Consistent with these human studies, obese mice exhibit innate airway hyperresponsiveness (AHR)[9], a cardinal feature of asthma. Obesity also affects response to asthma

[1]Program in Molecular and Integrative Physiological Sciences, Department of Environmental Health, Harvard T.H. Chan School of Public Health, Boston, MA 02115, USA. [2]Emergency Medicine, Beth Israel Deaconess Medical Center, Boston, MA 02115, USA. [3]Computational Health informatics Program, Boston Children's Hospital, Boston, MA 02115, USA. [4]Asthma Research Center, Brigham and Women's Hospital, Harvard Medical School, Boston, MA 02115, USA. [5]Department of Pediatrics and Department of Molecular and Integrative Physiology, University of Michigan Medical School, Ann Arbor, MI 48109, USA. [6]Channing Division of Network Medicine, Brigham and Women's Hospital, Harvard Medical School, Boston, MA 02115, USA. [7]Department of Biostatistics, Epidemiology and Informatics, University of Pennsylvania, Philadelphia, PA 19104, USA. [8]Division of Pediatric Respiratory Medicine, University of California San Diego and Rady Children's Hospital, San Diego, CA 92123, USA. ✉e-mail: qlu@hsph.harvard.edu

therapy[7]. For example, corticosteroids are less effective in asthma patients with obesity[10–13], for whom asthma control is more difficult to achieve[14–17]. There is also evidence that bronchodilators are less effective in asthma patients with obesity[18]. Despite the overwhelming evidence linking obesity to worsened asthma and to reduced response to therapy, the underlying mechanisms are not well understood. Consequently, there are few therapeutic options for asthma patients with obesity.

Cholecystokinin (CCK) is a peptide hormone found predominantly in the gastrointestinal (GI) tract and throughout the central nervous system (CNS)[19]. In the GI tract, CCK is released in response to food intake to regulate the motility and contraction of gall bladder and stomach[20,21]. In the nervous system, CCK acts as a neurotransmitter that signals satiation and modulates nociception[22]. While the function of CCK has been described mostly in the GI tract and in the CNS, CCK has also been shown to stimulate the secretion of calcitonin, insulin and glucagon and to act as a natriuretic kidney peptide[23]. CCK has two specific receptors, cholecystokinin A receptor (CCKAR) and cholecystokinin B receptor (CCKBR), which are expressed predominantly in the GI tract and nervous system to mediate the action of CCK[24]. In the GI tract, CCKAR is the predominant CCK receptor that induces the contraction of smooth muscle of gall bladder and stomach[25,26]. CCK receptors have been detected in the lung[27–30]. In isolated rat pulmonary interstitial macrophages, CCK receptors mediated the inhibitory effect of CCK on lipopolysaccharide-induced TNF-α and IL-1β mRNA expression[31,32]. Studies have also shown that CCK receptors are expressed by small cell lung cancer cells and that CCK elevates cytosolic calcium in these cells, suggesting a functional signaling pathway[28,33–35]. Intriguingly, Stretton and Barnes showed that a CCK octapeptide constricts both guinea pig and human airways[36]. However, the potential in vivo function of CCK in AHR or asthma was not further explored, and it was not known which CCK receptor mediates the CCK action in the airway.

In this study we present evidence that CCK and its receptor CCKAR are functionally expressed in and induce contraction of ASM cells. In vivo, CCK is upregulated in the airways of diet-induced and genetically obese mice, suggesting that the hormone contributes to the heightened AHR associated with obesity. Our data using obese mouse models further demonstrate that antagonizing CCK/CCKAR in the lung attenuates obesity-induced AHR. Our studies provide critical preclinical evidence that supports the development of CCKAR antagonists into new therapeutics to treat asthma patients with obesity.

## Results
### CCKAR expression in ASM
A large family of proteins known as G-protein-coupled receptors (GPCR) are direct targets of about 35% of all modern pharmaceuticals[37]. Many existing asthma drugs such as β-agonists, anti-cholinergics, and leukotrienes work by targeting their cognate GPCRs in the airways[9]. To identify potential new therapeutic targets for asthma, we evaluated the expression of non-olfactory GPCRs (361 in total) in primary human ASM cells using our previously published RNA-seq-based transcriptomic dataset[38]. We assessed the expression of each non-olfactory GPCR based on the values of Fragments Per Kilobase of transcript per Million (FPKM) mapped reads. From this analysis, we identified 115 (~32%) non-olfactory GPCRs that are moderately-to-highly expressed (FPKM > 0.1) in ASM cells (Supplementary Data).

Among the top 5 expressed GPCRs, CCKAR has not been previously implicated in any aspect of ASM function. *CCKAR* expression in ASM cells has a high FPKM value of 49.63 (Fig. 1a and Supplementary Data). In comparison, the β-adrenergic receptor 2 (gene name *ADRB2*), which is the target of bronchodilator β-agonists, has an FPKM value of 0.27 (Fig. 1a and Supplementary Data). Interestingly, the other CCK

receptor CCKBR is not expressed at all in the ASM cells (no reads in the RNA-seq data). The expression and function of CCKAR have been mostly investigated in extrapulmonary tissues such as the gallbladder and the sphincter of Oddi where it mediates CCK-induced contraction and relaxation, respectively[39,40]. However, CCKAR's surprisingly high expression in ASM cells suggests a potential biological role for the receptor in the ASM and in airways.

We next validated the expression of *CCKAR* in ASM cells using qRT-PCR analysis. We found that *CCKAR* is highly expressed in ASM cells but is very lowly expressed in other cell types of airways, including lung fibroblasts, basal and differentiated bronchial epithelial cells (Fig. 1b). Moreover, immunostaining of human airway tissue sections supported the expression of CCKAR in the lung (Fig. 1c and Supplementary Fig. S1), specifically in ASM as shown by colocalization with the known ASM marker α-smooth muscle actin (αSMA) (Fig. 1c). Together these results demonstrate the expression of CCKAR in ASM cells.

### Activation of CCKAR induces ASM contraction
CCKAR is known to mediate smooth muscle contraction in gallbladder and pyloric sphincter[39,40]; however, in the sphincter of Oddi, CCKAR mediates relaxation of the muscle tissue[39,40]. To test whether CCKAR evokes contraction or relaxation in ASM cells, we measured changes of cell stiffness using a biomechanical assay known as optical magnetic twisting cytometry (OMTC)[41]. Changes in cell stiffness measured in this way are tightly related to changes in contractile force[42]. Upon treatment with a highly specific CCKAR agonist A71623 (10 nM) the stiffness of ASM cells increased (Fig. 1d). Such increase in cell stiffness is comparable in magnitude and timing (~1.4 fold, ~50 s) to that induced by acetylcholine, a known ASM constrictor, and opposite to that induced by albuterol, a β-agonist that relaxes ASM cells. To assess more directly the effect of A71623 on ASM contractility, we used traction force microscopy (TFM), which measures the contractile forces exerted by cells on their surroundings (e.g. extracellular matrix)[43]. Using TFM, we measured contractile force in ASM cells pre- and post-treated with A71623 and found that A71623 increased ASM cell contraction by ~2.4 fold in the force response ratio compared to control (Fig. 1e, f). Moreover, A71623 enhanced acetylcholine-induced contraction of ASM cells as the effect of adding both compounds was greater than that induced by either compound alone (Fig. 1e, f). Together these biophysical measurements demonstrated that the CCKAR agonist A71623 induces contraction of ASM cells.

We next investigated whether ASM contraction by A71623 is specifically mediated by CCKAR. We first generated independent *CCKAR*-knockout ASM lines (*CCKAR*-KO1 and *CCKAR*-KO2) using the CRISPR/Cas9 technology with two different guide RNAs targeting distinct regions of the *CCKAR* gene (Fig. 2a). CRISPR-mediated knockout of *CCKAR* prevented A71623-induced contraction of ASM cells as measured by TFM (Fig. 2b). Similarly, there was no increase in cell stiffness upon A71623 treatment in the two *CCKAR*-KO ASM cell lines (Fig. 2c). Consistent with the results in *CCKAR*-KO cells, pre-treatment with a pan-CCK antagonist lorglumide (Fig. 2d) or a highly specific CCKAR antagonist devazepide (Fig. 2e) completely abolished A71623-induced stiffening of ASM cells. Together, these data demonstrate that CCKAR is required for CCK agonist-induced contraction in ASM cells.

### CCK expression and induction by free fatty acids in ASM cells
Our in vitro biophysical experiments so far have used a synthetic analogue of CCK hormone (A71623) that was given exogenously to ASM cell cultures. Studies have shown that cells that express CCK receptors often also express endogenous CCK, which provides evidence for autocrine activation and signaling of CCKAR[44,45]. We thus examined whether ASM cells also express CCK. Our qRT-PCR measurements (Fig. 3a) showed that *CCK* is expressed in ASM cells at a level

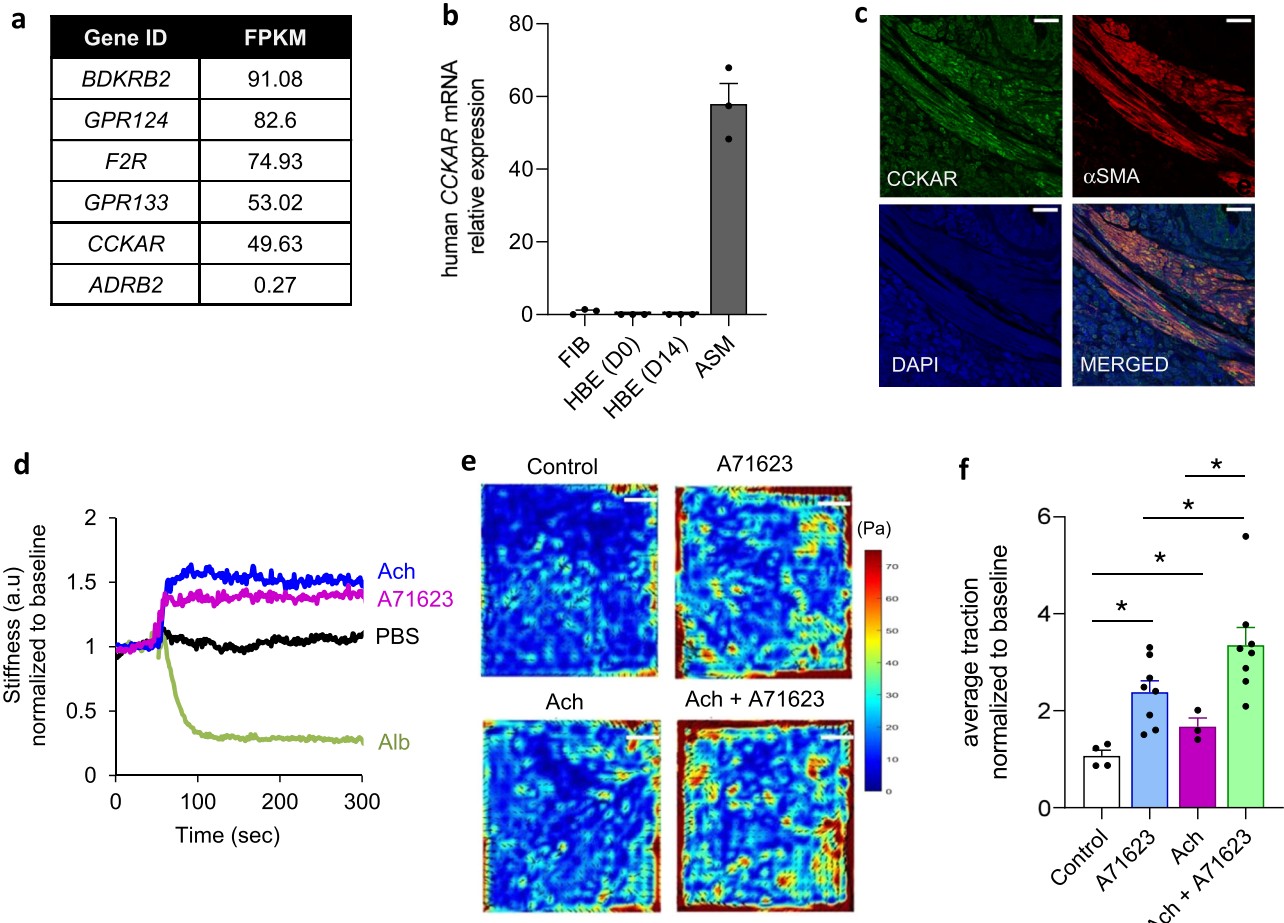

**Fig. 1 | ASM cells express CCKAR and contract in response to CCK agonist. a** List of the top non-olfactory GPCRs expressed in ASM cells. **b** qRT measurement of CCKAR mRNA expression in different airways cell types. FIB: primary lung fibroblasts; HBE: human bronchial epithelial (HBE) cells: HBE(D0) day 0 of initiation of air-liquid interface (ALI) culture; HBE(D14): day 14 post-ALI; ASM: airway smooth muscle cells. Data are presented as mean ± SEM. $n = 3$ biologically independent samples per group. **c** Immunostaining of human lung tissue section for CCKAR (green). α-SMA (α-smooth muscle actin, red) was used as a marker of ASM cells. DAPI was used to stain the nuclei (blue). Scale bar = 25 μm. Shown is a representative image from two independent experiments with similar results. **d** OMTC-based measurement of cell stiffness in ASM cells treated with CCK agonist A71623 (10 nM), acetylcholine (Ach), β-agonist (Alb: albuterol) or PBS (vehicle). a.u.: arbitrary unit. **e** Representative TFM images in ASM cells treated with vehicle (Control), A71623 (10 nM), acetylcholine (Ach, 1 μM) or combination of A71623 (10 nM) and Ach (1 μM). Scale bar = 10 mm). **f** Quantification of TFM-based contractility measurement in ASM cells treated with vehicle ($n = 4$), A71623 ($n = 8$), Ach ($n = 3$), or combination A71623 and Ach ($n = 8$). Data are presented as mean ± SEM; biologically independent samples per group; A71623 vs. Control ($P = 0.0020$); Ach vs. Control ($P = 0.0160$); A71623 vs Ach + A71623 ($P = 0.0222$); Ach vs. Ach + A71623 ($P = 0.0130$); one-tailed, unpaired $t$-test (*$P < 0.05$). Source data are provided as a Source Data file.

much higher than in other airway cells such as lung fibroblasts, basal airway epithelial cells, or differentiated airway epithelial cells. Immunostaining experiment also showed that CCK is expressed in the ASM cells (marked by colocalization with α-SMA) in the mouse airways (Fig. 3b).

As the ligand binding site of CCKAR is located extracellularly[46], we next tested whether endogenous CCK is secreted by ASM cells. In vivo, the CCK hormone is induced and secreted by cells lining the small intestine in response to ingestion of food, especially those with high fat content[47]. Because free fatty acids have been shown to induce CCK secretion in culture[47] and in vivo[48], we next sought to determine if CCK may likewise be induced by free fatty acids in ASM cells. We treated ASM cells with various types of free fatty acids and assessed their effects on CCK expression and secretion. As shown in Fig. 3c, treatment with long-chain free fatty acids (palmitoleic and myristoleic acids) increased CCK mRNA by ~3.5 fold in ASM cells, whereas treatment with free fatty acids with shorter carbon chains (hexanoic acid, decenoic acid, or dodecanoic acid) did not significantly increase CCK mRNA expression. Consistent with the mRNA

expression data, the CCK hormone level increased by ~30 and ~15 fold in ASM cells treated with palmitoleic and myristoleic acids, respectively, as detected by ELISA (Fig. 3d). Together these data show that expression of CCK hormone is induced by free fatty acids in ASM cells.

**Elevated CCK levels in the lungs of obese mice**

Because circulating free fatty acids are elevated in obesity[49,50], which is associated with the development of AHR[9], and because our data revealed that fatty acids induce CCK expression in ASM cells, we asked whether CCK expression is elevated concomitantly with an increased level of fatty acids in vivo. We examined the lungs of both high-fat-diet (HFD)-induced obese and genetically obese (*db/db*) mice. We found that the levels of free fatty acids (FFA) in both serum and lungs of HFD-fed obese mice were elevated compared to regular chow-fed lean mice (Supplementary Fig. S2a, b). Importantly, there were concomitant increases in both *CCK* mRNA and CCK protein levels in the lungs of HFD-fed mice compared to regular chow-fed mice (Fig. 3e, f). Similarly, the levels of FFA in the lungs and serum of *db/db* mice were elevated

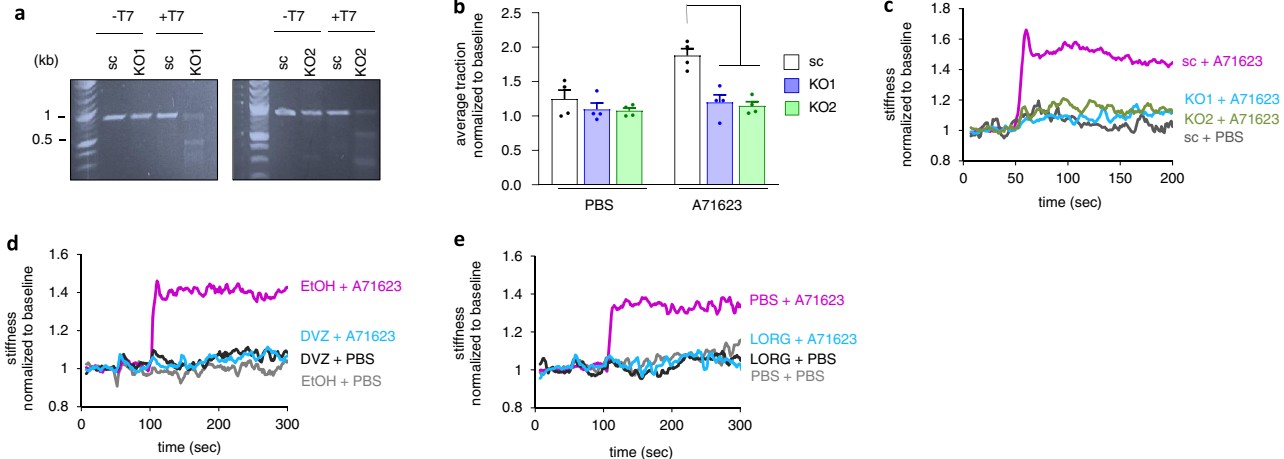

**Fig. 2 | CCKAR mediates CCK-induced stiffening and contraction of ASM cells.**
**a** T7E1 assay on CRISPR/Cas9-mediated *CCKAR* knockout ASM cells. *CCKAR* knockout (KO) cells were generated by lentiviral delivery of gRNAs targeting the *CCKAR* gene. DNA cleavage indicating mismatch and mutation induced by the two gRNAs targeting *CCKAR* was detected by T7 Endonuclease 1. Multiple bands in the DNA gel indicate mutation in the *CCKAR* gene in KO cells (KO1 and KO2). **b** TFM-based contractility measurement in control vs. *CCKAR*-KO ASM cells in response to A71623 treatment (10 nM). Data are presented as mean ± SEM; *n* = 4 biologically independent samples per group; sc vs. KO1 (*P* = 0.002); sc vs. KO2 (*P* = 0.0004); one-tailed, unpaired *t*-test (*P* < 0.05). **c** OMTC-based measurement of cell stiffness in control vs. *CCKAR*-KO cells in response to A71623 treatment (100 nM). **d**, **e** OMTC-based measurement of cell stiffness in response to A71623 treatment (100 nM) and in the presence of devazepide (**d**) or lorglumide (**e**). Ethanol and PBS were used as vehicles for devazepide and lorglumide, respectively. sc Scrambled control, dvz Devazepide, lorg Lorglumide, EtOH Ethanol, a.u. Arbitrary unit. Source data are provided as a Source Data file.

compared to wild type (WT) mice (Supplementary Fig. S2c, d), and the levels of *CCK* mRNA and protein were elevated in the lungs of *db/db* mice compared to WT mice (Fig. 3g, h). Together these data reveal elevated CCK levels in the lungs of obese mice.

## Antagonizing CCK/CCKAR attenuates the innate AHR in obese mice

Obese mice exhibit innate AHR, a clinical feature of asthma[9]. Our data showing elevated CCK in the lungs of obese mice suggest that a heightened CCK/CCKAR signaling and subsequent ASM contraction may contribute to the development of obesity-associated AHR. We therefore directly tested whether antagonizing CCKAR counteracts AHR in obese mouse models. Mice fed with high-fat diet (60% kcal/fat) gain weight very rapidly and develop AHR, which becomes pronounced at 21–24th weeks of age compared to the regular-chow-fed, lean controls[51]. We administered CCKAR antagonists into HFD-obese and control mice and then assessed AHR in response to increasing doses of methacholine (Fig. 4a). CCKAR antagonists were administered via intranasal delivery, which may avoid undesirable side effects and limit the gastrointestinal-hepatic first-pass metabolism that can contribute to loss of administered drugs[52,53]. Consistent with the known effects of HFD on airway function[51], in the absence of CCKAR antagonists, we observed an increased AHR in HFD-fed mice as compared to the regular chow mice (Fig. 4b, c). Importantly, intranasal administration of either proglumide (a pan-CCK antagonist) or devazepide (a potent and highly selective CCKAR antagonist) significantly reduced AHR in HFD-fed obese mice, as shown by the decrease in airway resistance ($R_L$) to a level similar to that of lean controls (Fig. 4b, c). The antagonist treatments did not affect the airway resistance in regular-chow-fed mice (Fig. 4b, c).

We likewise tested the effect of antagonizing CCKAR on AHR in genetically induced obese mice (Fig. 4d). *Db/db* mice, which lack the receptor for leptin, rapidly develop obesity on the standard chow diet and exhibit AHR starting at 8 weeks, even in the absence of any inciting exposure[9,54]. Consistent with the known effects of genetic obesity on airway function[9], in the absence of CCKAR antagonists, we observed an increased AHR in *db/db* mice as compared to the WT mice (Fig. 4e, f). Similar to the effect of antagonizing CCKAR in HFD-fed mice,

intranasal administration of either proglumide or devazepide significantly reduced AHR in the *db/db* obese mice (Fig. 4e, f).

Because inflammation may contribute to the development of AHR, we investigated whether blocking CCKAR affects airway inflammation. We found that the bronchoalveolar lavage (BAL) cell counts were not changed by CCKAR antagonist treatments in either HFD-fed (Supplementary Fig. S3a, b) or *db/db* obese mice (Supplementary Fig. S3c). We also assessed the levels of a panel of BAL inflammatory molecules/cytokines. We found that the majority of cytokines were not altered in obese mice or by CCKAR antagonists (Supplementary Fig. S4). We observed two cytokines (eotaxin and MIP-1α) that were increased in BAL of *db/db* mice. However, CCKAR antagonist treatment did not alter the levels of these two cytokines. Together these results suggest that antagonizing CCKAR attenuates the innate AHR in obese mice without affecting airway inflammation.

## Discussion

Obesity is a significant co-morbidity that associates with increased prevalence and severity of asthma[7]. The molecular mechanisms underlying the obesity-asthma association remain poorly understood. The lack of concrete knowledge of the pathophysiology of obesity-associated asthma precludes the development of new therapies that could effectively treat asthma in patients with obesity. Our study uncovers a previously undescribed mechanism for obesity-associated asthma that is based on the largely unexplored ability of CCK hormone and its receptor CCKAR in promoting ASM contraction. Furthermore, our study provides direct pre-clinical evidence for the repurposing of potent CCKAR antagonists as new, potentially effective therapies for asthma patients with obesity.

Although CCK increases airway constriction ex vivo[36], neither the identity of the cognate receptor nor the in vivo implication of such airway constriction was known. Our study first identified CCKAR as a highly expressed GPCR in ASM cells from an unbiased RNA sequencing dataset that led us to investigate its role in CCK-induced airway constriction. Using genetic ablation and pharmacologic intervention, we firmly established the role of CCK and CCKAR in contracting ASM cells. We also found that CCK is elevated in the lungs of mouse models of obesity-associated asthma, which implied that CCK/CCKAR mediates

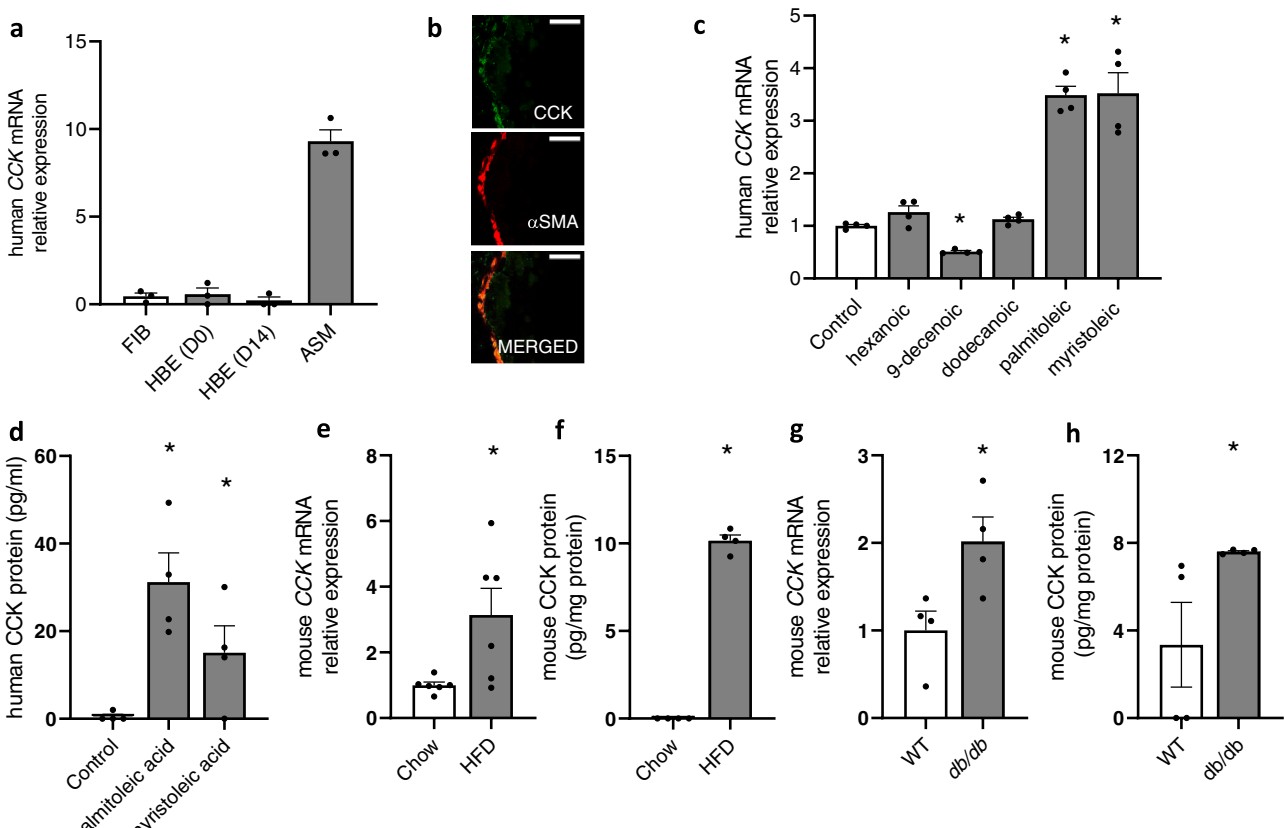

**Fig. 3 | CCK expression in ASM and in the lung. a** qRT measurement of CCK mRNA expression in airway cells. FIB: lung fibroblasts; HBE: human bronchial epithelia cells; HBE(D0): day 0 of initiation of air-liquid interface (ALI) culture; HBE(D14): day 14 post-ALI; ASM: airway smooth muscle cells. (n = 3 biologically independent samples per group). **b** Immunostaining of mouse lung tissue section for CCK (green). α-SMA (α-smooth muscle actin, red) was used as a marker of ASM. Scale bar = 25 μm. Shown is a representative image from three independent experiments with similar results. **c** qRT measurement of *CCK* mRNA expression in ASM cells after treatment with indicated free fatty acids (250 μM) or control (isopropanol). n = 4 biologically independent samples per group; hexanoic vs. control (P = 0.0772); 9-decenoic vs. control (P < 0.0001); dodecanoic vs. control (P = 0.0533); palmitoleic vs. control (P < 0.0001); myristoleic vs. control (P = 0.0007). Data represent the mean ± SEM; two-tailed, unpaired t-test (*P < 0.05). **d** ELISA measurement of CCK in ASM cell culture medium after free fatty acid treatment (n = 4 biologically

independent samples per group); palmitoleic acid vs. control (P = 0.0019); myristoleic acid vs. control (P = 0.0280). Data represent the mean ± SEM; one-tailed, unpaired t-test (*P < 0.05). **e** qRT measurement of CCK mRNA expression in the lungs of HFD-fed mice vs. regular chow-fed mice. n = 6 mice per group. Data represent the mean ± SEM; P = 0.0132; one-tailed, unpaired t-test (*P < 0.05). **f** CCK protein expression in the lungs of HFD-fed mice vs. regular chow-fed mice. n = 4 mice per group. Data represent the mean ± SEM; P < 0.0001; one-tailed, unpaired t-test (*P < 0.05). **g** qRT measurement of *CCK* mRNA expression in the airways (lungs and trachea) db/db mice compared to wild type control (WT). n = 4 mice per group; Data represent the mean ± SEM; P = 0.0152; one-tailed, unpaired t-test (*P < 0.05). **h** CCK protein expression in the lungs of db/db mice (n = 4) vs. wild-type control (WT). n = 4 mice per group; Data represent the mean ± SEM; P = 0.0353; one-tailed, unpaired t-test (*P < 0.05). Actin or 18 s rRNA was used as reference gene. Source data are provided as a Source Data file.

AHR in the obese. Indeed, blocking CCKAR using potent CCKAR antagonists attenuates obesity-associated AHR in two different obese mouse models. Our results support a model in which elevated CCK level associated with obesity increases CCKAR signaling in the lung to constrict ASM cells to drive bronchoconstriction and AHR. Antagonizing such elevated CCK/CCKAR signaling in the ASM blocks obesity-induced AHR and thus presents a completely new way to treat asthma in patients with obesity (Fig. 4g).

Our studies clearly established the in vivo role of CCK-induced airway constriction as blocking the heightened CCK/CCKAR signaling using potent antagonists decreased the pulmonary resistance to methacholine challenge in two distinct mouse models of obesity-associated asthma—a genetically obese mouse model and a diet-induced mouse model. CCKAR antagonists are intended for gastric ulcer to inhibit gastrointestinal motility and gastric secretions and are known to possess favorable safety profile[55,56]. Thus, our data showing that CCKAR antagonists attenuated AHR support the idea that these drugs could potentially be repurposed for the treatment of asthma patients with obesity. We used two antagonists (proglumide and devazepide) that possess different selectivity profiles with respect to

CCK receptors. Devazepide, in addition to being more potent in reducing AHR, is also the more selective CCKAR antagonist than the pan-CCK receptor-targeting proglumide. Thus, the use of devazepide or other highly selective CCKAR antagonists would be a priority for asthma drug repurposing to limit the undesirable effects brought by the possibility of blocking the other CCK receptor (CCKBR). CCKAR antagonists are normally administered via oral or intraperitoneal delivery that increases drug bioavailability in the systemic circulation. We showed that intranasal administration of CCKAR antagonists is sufficient to reduce AHR in obese mice. Through this delivery method, the potential unwanted effects of CCKAR inhibition in extrapulmonary tissues is likely limited. Potent bronchodilators are commonly delivered via nebulization, suggesting that CCKAR antagonists may be developed in similar formulation to effectively abrogate asthma symptoms while avoiding undesirable systemic effects.

Our data suggest that, at least in the mouse with obesity-associated AHR, the heightened CCK/CCKAR signaling in the lung appears to lead to increased ASM contraction that is independent of airway inflammation as CCKAR antagonists did not alter markers of inflammation. This is supported by our in vitro biophysical studies

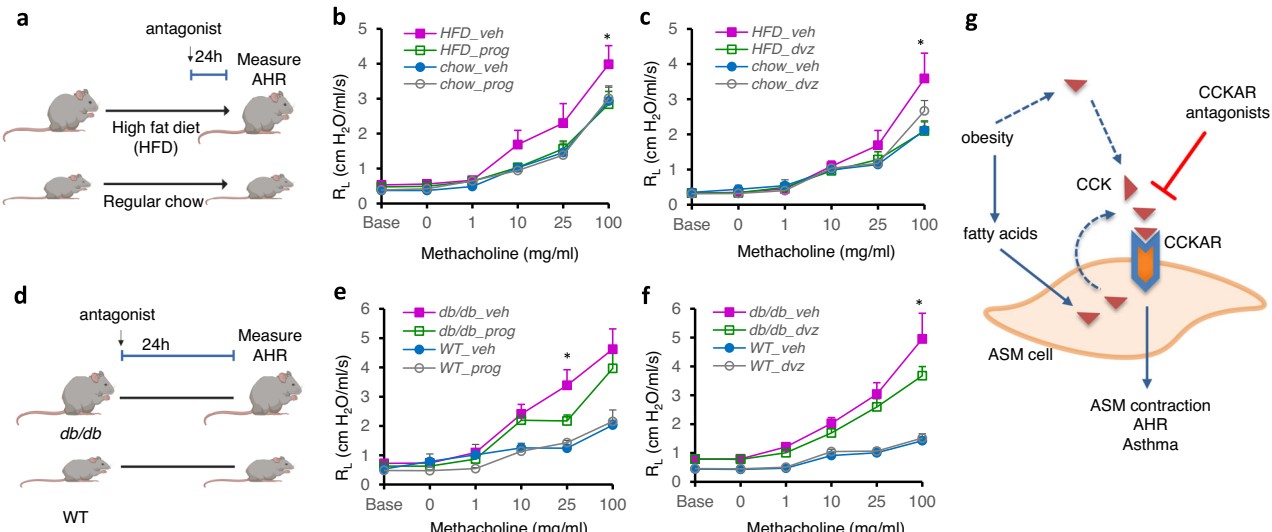

**Fig. 4 | CCK contributes to innate AHR in mouse models of obesity-associated AHR. a** Scheme of CCKAR antagonist treatment in HFD or regular chow-fed mice. Eight-week old C57BL/J6 mice were fed with HFD or regular chow for at least 13 weeks. Mice were then given proglumide (50 mg/kg) or devazepide (25 μg/kg) intranasally and AHR was assessed by Flexivent. **b** Pulmonary resistance in HFD-fed mice treated with proglumide (HFD_prog, n = 11) or vehicle (HFD_veh, n = 12) and regular chow-fed mice treated with proglumide (chow_veh, n = 8) or vehicle (chow_veh, n = 9). *P = 0.0070 (HFD_prog vs. HFD_veh); F values: treatment (F = 5.664, P = 0.0009); methacholine (F = 63.60, P < 0.0001); interaction (F = 0.7942, P = 0.6835). **c** Pulmonary resistance in HFD-fed mice treated with devazepide (HFD_dvz, n = 9) or vehicle (HFD_veh, n = 9) and regular chow-fed mice treated with devazepide (chow_dvz, n = 5) or vehicle (chow_veh, n = 6). *P < 0.0001 (HFD_dvz vs. HFD_veh); F values: treatment (F = 2.934, P = 0.0354); methacholine (F = 50.13, P < 0.0001); interaction (F = 1.645, P = 0.0684). **d** Scheme of CCKAR antagonist treatment in db/db or WT control mice. Mice were given proglumide (50 mg/kg) or devazepide (25 μg/kg) intranasally and AHR was assessed by Flexivent. **e** Pulmonary resistance in db/db mice treated with proglumide (db/db_prog, n = 8) or vehicle (db/db_veh, n = 9) and WT mice treated with proglumide (WT_prog, n = 5) or vehicle (WT_veh, n = 6). *P = 0.0184 (db/db_prog vs. db/db_veh); F values: treatment (F = 16.73, P < 0.0001); methacholine (F = 39.54, P < 0.0001); interaction (F = 2.993, P = 0.0003). **f** Pulmonary resistance in db/db mice treated with devazepide (db/db_dvz, n = 11) or vehicle (db/db_veh, n = 12) and WT mice treated with proglumide (WT_prog, n = 6) or vehicle (WT_veh, n = 6). *P = 0.0074 (db/db_dvz vs. db/db_veh); F values: treatment (F = 25.61, P < 0.0001); methacholine (F = 29.68, P < 0.0001); interaction (F = 3.544, P < 0.0001). **g** Possible mechanism of obesity-associated AHR. Obesity may induce CCK in ASM cells through circulating FFA or independent of FFA. This leads to secretion and binding of CCK to CCKAR on ASM cells, which in turn results in ASM contraction and AHR. Antagonizing CCKAR blocks CCK-induced ASM contraction and reduces obesity-associated AHR. Source data are provided as a Source Data file. Figure 4a and d were created by Biorender.com. Results represent the mean ± SEM. P and F values were determined using two-way ANOVA with Bonferroni correction.

showing that CCK causes direct ASM contraction. While our studies focused largely on inhibiting CCKAR-mediated contraction, the absence of effect of CCKAR antagonists on BAL cell differential counts or inflammatory cytokines nevertheless suggests that blocking CCKAR-mediated ASM contraction is sufficient to suppress obesity-associated AHR.

Our study focused on the expression and function of CCKAR in ASM cells, which are the main effector of bronchoconstriction, but previous studies had reported the expression of CCK receptors in other lung cell types, including bronchial epithelial cells, alveolar epithelial cells, pulmonary macrophages, and pulmonary vascular endothelial cells[29,30,32], whose roles in asthma are tightly linked to inflammation. Epithelial cells are the airway's first line of defense against inhaled allergens and pathogens, and epithelial damage initiates airway inflammation[57]; pulmonary macrophages are dysfunctional in allergic asthma and are major sources of TNF-α, IL-1β, IL-6 and IL-8[58]; and vascular endothelial cells may contribute to airway inflammation and remodeling through eotaxin, IL-1α, GM-CSF, and VEGF[59]. In pulmonary macrophages, CCK was shown to exert anti-inflammatory activity in response to endotoxin shock inducer, lipopolysaccharide[31,32]. As we did not observe significant changes in airway inflammation in the obese mouse models as indicated by the absence of significant differences in the levels of inflammatory molecules/cytokines as well as in percentage of macrophages in BAL, it is unlikely that the reduction in AHR by CCKAR antagonists was linked to the reported function of CCK in lung macrophages. Other studies also reported a role for CCK in promoting cell proliferation of lung cancer cells. However, as our dose regimen is considered acute treatment, it is also unlikely that the action of CCKAR

antagonists in reducing AHR in our mouse models was through inhibition of proliferation of asthma-relevant cells such as bronchial epithelial cells. Thus, while CCK receptors are also expressed in non-ASM cells in which they mediate cellular processes in the context of other lung disorders, CCKAR signaling in ASM cells appears to be the major driver of obesity-associated AHR.

Our findings suggest an autocrine stimulatory loop of CCK-activated, CCKAR-mediated ASM contraction where free fatty acids induce the expression and secretion of CCK in ASM cells. However, our studies have some limitations. Our experimental results cannot rule out the possibility that CCK in the GI and in the CNS also contribute to the AHR in the obese mice. Although our data suggests that CCK is produced in the lungs as shown by our qRT-PCR, immunostaining, and ELISA data, GI-produced CCK that has entered the circulation and reached the lungs could potentially activate CCKAR signaling in ASM. Plasma levels of CCK are elevated in obese rodents or after high-fat diet intake[60,61]. Individuals with high circulating CCK levels may thus be prone to altered lung function and heightened AHR. Similarly, conditions that lead to elevation of plasma CCK may trigger asthma symptoms. It was reported that exercise elevates plasma CCK levels in individuals with exercise-induced asthma[62]. Similarly, plasma CCK was observed to be increased by high altitude[63]. It is possible that the worsening of asthma symptoms associated with these clinical scenarios is related to elevated plasma CCK levels. Our results, however, do not identify which CCK source (circulating or ASM-secreted) contributes more greatly to obesity-induced AHR. It is likely that both CCK sources, independently or synergistically, lead to the contraction of ASM once CCKAR is engaged. Similarly, circulating FFAs may initiate ASM contraction

when they induce the production and the secretion of CCK. Regardless of the initiating event, the resulting AHR may be alleviated by blocking CCK/CCKAR signaling with the antagonists.

Our experimental results also do not rule out the possibility that CCKAR signaling in the CNS affects obesity-associated AHR. It has been shown that metabolic hormones can regulate AHR in obese mice by acting on cholinergic neurons in the brainstem[64,65] though this activity has not been reported for CCK. Although CCK levels are elevated in obese rodents or after high-fat diet intake[60,61], CCK itself is not able to cross the blood-brain barrier[66,67]. Its potential action in the CNS-mediated control of obesity-associated AHR, if at all, would likely have to be through brainstem-derived CCK. However, previously published reports of the level of CCK in the brain of obese mice *(ob/ob)* showed either no significant difference or lower compared to lean controls[68–70]. Furthermore, another published study showed that the level of binding of CCK to its receptors in the brain of *db/db* mice is unchanged[71], suggesting that there is no increased CCKAR signaling in the brain that could have contributed to the increased AHR of the *db/db* mice that we observed in our study. Tissue-specific conditional CCK or CCKAR knockout in obese mice may be needed to completely rule out the possible role of CCK/CCKAR in the brainstem in modulating obesity-associated AHR.

In summary, there is an urgent need to develop new asthma therapeutics especially for asthma patients with obesity. Our present study reveals that CCKAR mediates CCK-induced contraction of ASM cells and that CCK/CCKAR signaling is a mechanism underlying obesity-induced AHR in mice. Such findings imply that blocking CCKAR may be developed into a new asthma therapy for asthma patients with obesity, who still rely on ineffective symptomatic treatments. Our study provides critical pre-clinical evidence for repurposing CCK/CCKAR antagonists, many of them already developed for gastrointestinal diseases[61], to treat obesity-associated asthma. Future clinical studies built upon our findings here may lead to the development of much-needed effective therapies for asthma patients with obesity.

## Methods

### Ethics statement
The animal experiments were approved by the Harvard Medical Area Institutional Animal Care and Use Committee (HMA-IACUC) under the protocol #IS506-6. Primary ASM cells were isolated from lung tissues obtained from the National Disease Resource Interchange (NDRI). Their use was approved by the University of Pennsylvania Institutional Review Board. Use of these de-identified cells does not constitute human research.

### Cell culture and reagents
Primary cultures of human ASM cells were cultured as described[72]. ASM cultures were maintained in F12 (HAM) nutrient medium (Gibco) supplemented with 10% FBS, 100 units/ml penicillin, 100 µg/ml streptomycin, 300 µg/ml L-glutamine, 4.8 mM HEPES, 0.34 mM $CaCl_2$, and 2.4 mM NaOH. Passages 3–8 were used in the experiments.

Air-liquid interface (ALI) culture of normal human bronchial epithelial (HBE) cells (Cat. CC-2541, Lonza) was established and maintained until use[73]. HBE cells were first seeded in collagen-coated transwell inserts (0.4 µm pore size, Corning) and cultured under submerged conditions in a 1:1 mixture of bronchial epithelial basal media (Lonza) and DMEM (Mediatech, Tewksbury, MA) supplemented with bovine pituitary extract (52 mg/mL), hydrocortisone (0.5 mg/mL), human epidermal growth factor (0.5 ng/mL), epinephrine (0.5 mg/mL), insulin (5 mg/mL), triiodothyronine (6.5 ng/mL), transferrin (10 mg/mL), gentamicin (50 mg/mL), amphotericin-B (50 ng/mL), BSA (1.5 mg/mL), and retinoic acid (50 nM). When the cells reached confluence, the medium was removed from the apical surface. Air-liquid interface (ALI) culture was maintained up to at least 14 days.

Reagents used in the experiments included: A71623 (Cat. 2411), proglumide (Cat. 1478) and devazepide (Cat. 2304) from Tocris (Minneapolis, MN); lorglumide (Cat. L109, Sigma-Aldrich (St. Louis, MO); normal saline (Cat. S5819, Teknova, Hollister, CA); acetylcholine chloride (Cat. A6625), Tween-80 (Cat. P4780), DMSO (Cat D2650), hexanoic acid (Cat. W255912), 9-decenoic acid (Cat. W366005), myristoleic acid (Cat. M3525) and palmitoleic acid (Cat 76169) from Sigma-Aldrich (St. Louis, MO).

### Quantitative real-time (qRT)PCR
Total RNA was extracted using the RNEasy kit according to the manufacturer's instructions (Qiagen, Cat. 74106). RNA was then reverse-transcribed using the Superscript III First-Strand Synthesis System (Life Technologies, Cat. 18080-051). Quantitative PCR was performed using SYBR green master mix (Qiagen, Cat. 214145) with the following gene-specific primers synthesized by Integrated DNA Technologies (Coralville, IA):

Human *CCKAR* forward: 5′- TGCTCAAGGATTTCATCTTCG
Human *CCKAR* reverse: 5′- TGGTCACCAGATTAAAGGTAGATACA
Human *CCK* forward: 5′- CTGGCAAGATACATCCAGCA
Human *CCK* reverse: 5′- CCATGTAGTCCCGGTCACTT
Human *ACTB* (β-actin) forward: 5′- CCAACCGCGAGAAGATGA
Human *ACTB* (β-actin) reverse: 5′- CCAGAGGCGTACAGGGATAG
Human 18 s rRNA forward: 5′- CCGATTGGATGGTTTAGTGAG
Human 18 s rRNA reverse: 5′−AGTTCGACCGTCTTCTCAGC
Mouse *CCK* forward: 5′- GCTGATTTCCCCATCCAAA
Mouse *CCK* reverse: 5′- GCTTCTGCAGGGACTACCG
Mouse *CCKAR* forward: 5′- GATGCCAGCCAGAAGAAATC
Mouse *CCKAR* reverse: 5′- ACAGCCATCGCTATCCTCAT
Mouse *ACTB* (β-actin) forward: 5′- CTAAGGCCAACCGTGAAAAG
Mouse *ACTB* (β-actin) reverse: 5′- ACCAGAGGCATACAGGGACA

### Immunofluorescence staining
Immunofluorescence staining of lung tissue sections were performed in paraffin-embedded lung tissue[53]. The tissue slides were deparaffinized using xylene and rehydrated by washing with 100% ethanol, 95% ethanol, 70% ethanol, and finally distilled water. Antigen retrieval was performed by bringing slides to a boil in 10 mM citrate buffer, pH 6.0, then matinining at a sub-boiling temparture for 10 min. After cooling sildes for 30 min, the slides were washed with PBS. Lung tissue sections were blocked with PBS supplemented with 5% normal donkey serum and 0.2% Triton X-100 for 1 h at room temperature. Primary antibodies used were anti-CCKAR (Pierce, Cat. PA3-116; 1:500 dilution), anti-CCK (Santa Cruz Biotechnology, Inc., Cat. Sc-21617; 1:25 dilution), and anti-α-SMA (Sigma, Cat. C6198; 1:200 dilution). For isotype controls, normal rabbit whole serum (ThermoScientific, Cat. 31883) and normal goat IgG (Sta. Cruz, Cat. Sc-2028) were used. Secondary antibodies used were AlexaFlour 488 anti-rabbit (Life Technologies, Cat. A21206, 1:200 dilution) and anti-goat CF™488A (Sigma-Aldrich, Cat. SAB4600032, 1:200 dilution). ProLong gold anti-fade mountant with DAPI (Life Technologies, Cat. P36941) was used for mounting samples. Confocal images were taken using Leica SP8X Confocal Microscope and the images were processed using ImageJ.

### Generation of CCKAR-knockout cells
Two CCKAR-knockout cell lines (KO1 and KO2) were generated using CRISPR with the following guide RNAs (gRNAs): TAGAAAGCGGCA-CATATTCG for KO1 and CCGCTTGTTCCGAATCAGCA for KO2. Guides were cloned into lentiCRISPRv2 vector containing hSpCas9 cassette (Addgene)[54]. Oligonucleotides targeting the site sequence were synthesized with 3-bp NGG PAM sequence flanking the 3′ end, annealed and cloned into the BsmBI-digested lentiCRISPRv2 vector. The resulting plasmids were transformed into Stbl3 bacteria (ThermoFisher Scientific, Cat. C737303) and purified using Mini-prep Kit (Qiagen, Cat. 27104). Lentiviruses were produced by co-transfecting the

lentiCRISPRv2 containing gRNAs with the packaging plasmids pVSVg and psPAX2 (Addgene) in HEK293T cells (Cat. CRL-3216, American Type Culture Collection, ATCC). Lentiviral transduction in ASM cells was performed in the presence of polybrene (8 μg/ul) for 24 hours. Stably transduced ASM cells were selected using puromycin (ThermoFisher Scientific, Cat. A1113803). T7E1 assay was performed to determine knockout efficiency. For the T7E1 assay, the genomic region harboring the target of gRNAs was first PCR amplified, subjected to denaturing and reannealing temperatures (95 °C for 2 min, ramp down at −2 °C/s to 85 °C, ramp down at −0.1 °C/s to 25 °C, and stopped at 16 °C. T7E1 (New England Biolabs, Cat. M0302S) cleavage reaction was then performed at 37 °C for 20 min. The PCR products were visualized using 1.5% agarose gel.

## Optical magnetic twisting cytometry

Cell stiffness was measured using optical magnetic twisting cytometry[41]. ASM cells were cultured to confluence in collagen-coated 96-well plate (Stripwell Microplates, Corning, NY). After overnight serum starvation, cells were incubated with ferrimagnetic beads (4.5 μm in diameter, produced in house, and coated with poly-L-lysine) for 20 min to allow beads to attach to cell surfaces[74]. The beads were then magnetized with a strong magnetic pulse in the horizontal direction and twisted with a much weaker oscillatory magnetic field (0.77 Hz) in the vertical direction. The ratio of magnetic torque to bead motion[41,74] was used to determine cell stiffness; values are expressed in arbitrary units (a.u.). To determine relative changes in cell stiffness after drug treatment, the average of stiffness normalized to baseline was calculated.

## Traction force microscopy

Cell contraction was measured using traction force microscopy[43]. Glass bottom 96-well plates containing custom elastic polyacrylamide substrate were prepared using a multi-layered approach. The plates were treated with 6 N NaOH for 1 h followed by silane solution for additional 1 h. Fluorescent bead solution (1 μm carboxylate-modified microspheres, Invitrogen) was added to the wells and allowed to air-dry overnight. Glutaraldehyde (0.25% in PBS) was then added for 30 min followed by washing. After drying, acrylamide gels (5.5% acrylamide, 0.076% bisacrylamide, Young's modulus = 1.8 kPa, thickness = 200 μm) were cast using a custom-made gel caster (Matrigen Life Technologies, CA). Gel surfaces were functionalized using sulfo-SANPAH (sulfosuccinimidyl-6-[4′-azido-2′-nitrophenylamino]hexanoate, 0.2 mg/ml) and coated with 0.2 μm sulfate microspheres (Invitrogen). Collagen (40 μg/ml) was then added to coat plates prior to seeding of ASM cells. ASM cells were cultured to confluence in the plates using the regular FBS-containing ASM medium followed by switching to serum free medium containing insulin (5.7 μg/ml) and transferrin (5 μg/ml). The cells were maintained for additional 48 h in this medium until the time of experimentation (e.g., drug treatment). For the traction microscopy procedure, a set of flourescence images per well were obtained using an inverted microscope (DMI 6000B, Leica Inc): before plating the cells (*reference*), prior to drug exposure (*baseline*), and 1 h after drug exposure (*treatment*). The cell-exerted displacement field was determined by comparing the *baseline* or *treatment* image to the *reference* image. The contractile force (per unit area) was computed from the obtained displacement field using Fourier-transform traction microscopy. The average contractile force was then determined by computing the root mean squared value from each force map. Finally, the force-response ratio of treatment vs. baseline contraction was calculated.

## Mouse models of obesity-associated AHR and drug treatment

Obese mice and their corresponding lean controls were acquired from the Jackson Laboratory and were allowed at least one week for acclimation prior to experimentation. A maximum of 4 mice per cage were permitted and animals were checked at least weekly after arrival.

Standard monitoring practices were applied for animals including monitoring for persistent recumbence, intractable pain, severe weight loss, tumor, severe central nervous system signs, dyspnea and cyanosis, prolapse of the penis or rectum, limb or spinal fractures, and dystocia. Water and enrichment (nestlet) were also provided in the cages (static caging system). The temperature in the animal room was maintained at 70 F and relative humidity at 50%. Light and dark cycle was from 7:00 am to 7:00 pm.

High fat diet (HFD)-fed mice were obtained from Jackson Laboratory; these were male C57BL/6 J mice placed on a high fat diet where 60% of calories were derived from fat in the form of lard (D12492, Research Diets) starting at 8 weeks until the time of experiments. Mice placed on a regular chow diet (PicoLab 5058, Lab Diets) served as corresponding lean controls. Mice were 21–24 weeks of age at the time of experiments. Mice fed with HFD gain weight very rapidly and develop AHR compared to their corresponding lean controls, which have normal airway reactivity[54]. *Db/db* mice (B6.BKS(D)-*Lepr*[db]/J), which lack the long form of the leptin receptor, were used as the mouse model of genetic obesity. *Db/db* mice are substantially obese compared to the age- and sex-matched WT controls (C57BL/6 J) and exhibit increased airway responsiveness to methacholine even in the absence of any inciting stimuli (e.g., ovalbumin challenge)[9]. *Db/db* and WT mice (male or female) were 10–12 weeks of age at the time of experiments.

Mice were treated via intranasal administration of CCKAR antagonists (proglumide or devazepide). Proglumide was administered at 50 mg/kg (3x) and devazepide at 25 μg/kg (2x) for a total volume of 25 μl (12.5 μl in each nostril). The vehicles used to deliver the antagonists are normal saline for proglumide and 0.05% DMSO/ 0.05% Tween-80/normal saline for devazepide. After experiments, mice were euthanized with an overdose of sodium pentobarbital (200 mg/kg i.p.).

## Measurement of Airway Hyperresponsiveness (AHR)

This study was approved by the Harvard Medical Area Standing Committee on Animals. The forced-oscillation technique was used to assess AHR using the Flexivent machine (flexiWare7, Scireq, Montreal, QC, Canada) which was conducted at least two hours after the last administration of CCKAR antagonists or vehicle. Mice were first anesthetized with xylazine (7 mg/kg) and sodium pentobarbital (50 mg/kg) and an incision along the tracheal wall was made to expose the trachea. The trachea was then cannulated with a tubing adaptor and then connected to the machine. The mice were ventilated and instrumented for the measurement of pulmonary mechanics and airway responsiveness in response to increasing doses of methacholine[75]. The chest wall was opened to expose the lungs to atmosphere pressure and a positive end expiratory pressure of 3 cm $H_2O$ was applied. Obese mice were ventilated at a respiratory rate of 180 breaths/minute while lean mice were ventilated at 150 breaths/min. Changes in pulmonary resistance ($R_L$) were assessed at the baseline and after delivery of aerosolized PBS and aerosolized methacholine in increasing doses.

## CCK enzyme immunoassay

CCK Enzyme Immunoassay Kit (Cat. RAB0039 St. Louis, MO) was used to detect CCK in samples. This assay is based on the principle of competitive enzyme immunoassay. Lung tissue lysates were prepared using RIPA buffer (Thermo Scientific, Waltham, MA) supplemented with protease and phosphatase inhibitor cocktails (Roche, Mannheim Germany) and protein concentration was measured using BCA assay (Thermo Scientific, Waltham, MA). Standards and equal amounts of samples were mixed with biotinylated CCK (final concentration was 20 pg/mL in every sample). A 96-well plate was coated with anti-CCK antibody and incubated overnight at 4 °C with gentle shaking (1–2 cycles/sec). Following incubation, the wells were washed 5 times with wash buffer. Samples and standards were added into appropriate wells. A blank well-containing assay diluent was also included. The

plate was incubated overnight at 4 °C with gentle shaking (1–2 cycles/sec). Following incubation, the wells were washed 5 times with wash buffer. A solution of HRP-streptavidin was added to each well and the plate was incubated for 45 min with gentle shaking at room temperature. The solution was discarded and the plate washed 5 times with wash buffer. TMB One-Step Substrate Reagent was then added to each well and the plate was incubated for 30 min at room temperature in the dark with gentle shaking. Following incubation, a stop solution was added to each well and absorbance was read at 450 nm (SpectraMax 190 Microplate Reader, Molecular Devices). A four-parameter logistic regression model was then used to plot the standard curve and to determine the amount of CCK in the samples.

### Free Fatty Acid Quantification

For the detection of FFA in samples, the Free Fatty Acid Assay Kit was used (Cat. ab65341, Abcam, Waltham, MA). For lung samples, FFAs were extracted using Lipid Extraction Kit (Cat. Ab211044, Abcam, Waltham, MA) from equal weight of lung tissue samples (40 mg). Serum samples were directly assayed. The FFA Assay Kit converts fatty acids into their CoA-derivatives which are then oxidized with the concomitant generation of color or fluoresence. Fifty (50) µl of samples (dissolved in supplied assay buffer if needed) were added to each well in a 96-well plate. The acyl-CoA synthesis was then performed by adding 2 µl of ACS reagent into the wells, mixed, and incubated at 37 °C for 30 min. A reaction mix (44 µl assay buffer, 2 µl fatty acid probe, 2 µl enzyme mix, and 2 µl enhancer was added to each well and incubated at 37 °C for 30 min in the dark. After incubation the absorbance was measured at 570 nm (SpectraMax 190 Microplate Reader, Molecular Devices). A standard curve was generated using palmitic acid and the concentration of fatty acid was determined using formula, [FFA] = Fa/Sv (nmol/µl or mM) where Fa is the fatty acid amount in the well obtained from the standard curve and Sv is the sample volume (µl) added to the sample well. For lung samples, the amount of FFA was expressed per weight (g) of lung.

### Assessment of airway inflammation

Airway inflammation was assessed and compared between experimental groups using differential cell analysis and measurement of BAL cytokine levels. Lavage was performed in situ with a 20-gauge catheter inserted into the proximal trachea, flushing the lungs 4 times with 0.8 ml PBS. The recovered fluid of the 1st lavage was transferred into a 1.5 ml Eppendorf tube on ice, while the 3 subsequent lavages were pooled into a 5 ml Eppendorf tube on ice. BAL cells were separated from BAL fluid by centrifugation at 500 x g for 10 min at 4 °C. The BAL fluid of the 1st lavage was centrifuged again at 3000 x g for 10 min at 4 °C to spin down cell debris, the supernatant was frozen at −80 °C for further analysis of cytokines in the BAL. For total cell count, BAL cells retrieved from all 4 lavages were pooled and resuspended in 400 µl PBS, then 10 µl of the cell suspension was diluted 4-fold with trypan blue stain and manually counted twice using a hemocytometer. For differential count, a fraction of BAL cell suspension was cytospun onto microscopic slides, air-dried and stained with Diff-Quick. Four hundred (400) cells per sample were counted for the percentages of macrophages, eosinophils, neutrophils, and lymphocytes. For cytokine level analysis, a multiplex assay was performed on BAL samples using Mouse Cytokine 32-Plex Discovery Assay (Eve Technologies, Calgary, AB). Cytokines whose levels were below the detection limit were not included in the supplementary figures.

### Statistical analysis

Differences between groups were analyzed using Prism v8; GraphPad Software Inc. No statistical method was used to predetermine sample size and the investigators were not blinded to allocation during experiments and outcome assessment. For both cell culture and animal experiments, random assignment was used to assign samples and subjects to different treatment groups. Both F and P-values were reported for individual variables and their interactions, when appropriate. A $P < 0.05$ was considered statistically significant. Statistical tests used were noted in figure legends.

### Reporting summary

Further information on research design is available in the Nature Portfolio Reporting Summary linked to this article.

## Data availability

The data in Fig. 1a and Supplementary Data were generated from a previously published RNA-seq data available at the Gene Expression Omnibus Web site (http://www.ncbi.nlm.nih.gov/geo/) under accession GSE52778. All data supporting the findings described in this manuscript are available in the article and in the Supplementary Information and from the corresponding author upon request. Source data are provided with this paper.

## Code availability

The analysis codes for this study are available from the corresponding author.

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

## Acknowledgements

This work was supported mainly by a scholar award from the American Asthma Foundation and in part by a grant from the National Institutes of Health (R01HL139496) to Q.L. We thank Dr. Reynold Panettieri for providing human ASM cells.

## Author contributions

R.A.P. and Q.L. conceived the project and designed the experiments. R.A.P. performed most of the experiments. Q.L. supervised the project. C.Y.P, N.S., R.K., and J.F. performed the biomechanical assays and analyses. Z.Y., M.S., D.K., and S.A.S. contributed to AHR assessment in mice. E.I. and M.B.H contributed human samples for immunostaining. B.H., A.K., S.T.W., and K.G.T. supervised the RNA-Seq analyses. R.A.P. and Q.L. wrote the paper with input from other co-authors.

## Competing interests

Q.L. is a co-founder, scientific advisor and shareholder of Vesigen Therapeutics. All other co-authors declare no competing interests.
