## [Peer Review File · Nature Communications]

Antagonizing cholecystokinin A receptor in the lung attenuates obesity-induced airway hyperresponsivenessREVIEWER COMMENTS

Reviewer #1 (Remarks to the Author):

This is a 4 Figure manuscript submitted to Nature Communications investigating the role of CCK in obesity induced asthma. Authors use a combination of methods including genetic ablation and pharmacological intervention to show that the CCK/CCKAR axis contracts ASM cells in an autocrine manner, in a FFA-dependent mechanism. Authors further found that CCK is elevated in the lungs of mouse models of obesity-associated asthma, mediating AHR. The paper reads well. My major concern is the lack of immunology in the study, since authors conclude that CCK contracts ASM independently of the lung microenvironment (including immune cells). Please find my comments below.

Major comments.

- 1.) The big picture of the manuscript is the fact that CCK contracts ASM and induces AHR. This is why only 24 hours of treatment (Figure 4) is required to abrogate AHR using antagonists. However, obesity-induced asthma is mainly driven by the immune system. Authors try to look at TH17 but only say that there is no difference and the data is not shown. This reviewer believes that authors should study the immune system of these mice. What are the effects of the treatments on the immune system? Have the authors looked at inflammatory cells in the BAL? Hypothetically, CCK acts independently of the immune system, but authors should show it. This topic is briefly discussed in the discussion, but authors should show data in order to draw this conclusion.
- 2.) Although previously published, it would help the reader if authors show some of the RNAseq data in Figure 1 which led to focus on CCK.
- 3.) The text describing Figure 1 is lacking details. It would help the reader if authors describe the experiment that was performed. For example, what is d0, d14, HBE in Figure 1a? The order of graphs is misleading in Figure 1 and 2.
- 4.) In Figure 1c, Acetylcholine is used as a positive control. Would A71623 enhance the stiffness caused by Acetylcholine?
- 5.) Related to comment 3.). All of the CCK data is based on RNAseq or qPCR except Figure 3d which is an ELISA. What is the rationale for combining these readouts? Authors should comment.
- 6.) Authors discuss the role of circulating CCK in the discussion. However they do not put it in relation to their findings. What is the role of circulating CCK in their model? Is it high? Is the autocrine role of CCK in ASM more important? Are circulating CCK and FFA both involved? Authors should clarify.
- 7.) In Figure 4, some controls are missing. Chow/vehicle should be in all plots to show the basal levels of lung resistance. In addition, how did mice react to 100 mg/mL methacholine in Figure 4e? This is important since all other plots show up to 100 mg/mL methacholine.

Minor comments.

Line 31. Twice "for obsess asthmatics"

It is surprising to see a graphical summary in Figure 4 of a Nature communications manuscript

The ticks and methacholine doses are not in line in Figure 4

What is "Base" versus "0" in Figure 4?

Fig 3b missing scale bar

Reviewer #2 (Remarks to the Author):

In this manuscript the authors demonstrate a novel therapeutic target for treating obese asthmatics via antagonizing cholecystikinin (CCK) and its receptor CCKAR signaling in the airway smooth muscle cells. They first verified that CCK and CCKAR are highly expressed in airway smooth muscle cells but not in other types of airway residential cells such as fibroblasts and epithelial cells. Then, they demonstrate that, in both in vitro cell culture and in vivo mouse models, induction of CCK and activation of CCKAR led to increased contractility of airway smooth muscle cells in vitro and airway hyperresponsiveness (AHR) in vivo, which are all reversed by antagonizing CCKAR with known drug agents. These data seem to be very convincing that CCK/CCKAR antagonists may bring a great hope to those suffering from obesity-associated asthma because that this subtype of asthma is notorious for poor response to currently available therapy.

Overall, this is a well thought out and carefully designed study. The experimental coverage is quite thorough and the results appear to be technically sound with well-recognized techniques. Particularly in regarding the assays of ASMC stiffness and contraction using optical magnetic twisting cytometry (OMTC) and traction force microscopy (TFM), these results are very robust and trustworthy, which is not surprising because these two techniques are actually originated from one of the author's affiliated labs. Moreover, these two techniques have now been widely used in many research groups throughout the world and in many studies of airway smooth muscle cell mechanics in relation to diseases including but not limited to asthma. In this reviewer's eyes, the data of ASMC response in stiffness and contraction force to CCK/CCKAR stimulation are consistent and comparable to previously reported ASMC responses to other contractile agonists in the literature. There are only a couple of minor issues related to the OMTC protocol used in this study that may need the authors to clarify. 1) In this study, the beads were coated with non-specific binding poly-L-lysine instead of the more commonly used, specific integrin binding RGD peptide. What is the reason to change the surface coating agent for the magnetic beads? 2) the twisting frequency of magnetic field was set at 0.77 Hz, instead of usually used 0.3-0.5 Hz to mimic normal breathing rate. Any particular consideration of this setting?

The writing of the manuscript is in general of high quality with clarity and professionalism, except for a couple of minor typos such as 1) at the end of line 123, the "of" is redundant, 2) on line 294, "ALI" needs full spelling since it first appeared here, 3) on line 457, it should include "LORD: lorglumide", 4) once abbreviation of "ASM" was defined, it is better to replace all occurrences of "airway smooth muscle" afterwards throughout the manuscript.

Reviewer #3 (Remarks to the Author):

Panganiban and colleagues report a tidy and logical series of experiments that demonstrate that cholecystikinin (CCK) and its receptor CCKR1 is increased in the lungs of obese mice and that pharmacological blockade of CCK signalling reduce obesity-associated airway hyperresponsiveness (AHR). More specifically, authors suggest that CCK and CCKR1 are expressed by smooth muscle cells and regulate their contractility via a autocrine mechanism. The manuscript is interesting and overall easy to read. Below are some specific points that Authors might consider addressing and/or clarifying.

The abstract contains a few typos and repetitions.

Strictly speaking CCK it is known as a station signal, with minimal, if any, effect on satiety. Authors should consider a more rigorous description of CCK and its actions.

Also, it is suggested that Authors leverage on more recent references. For example, the statement "In the CNS, CCK acts as a neurotransmitter to regulate satiety and nociception" is not only limited, but also supported by a review published 40 years ago. Likewise, the segregation of CCK1 and CCK2 receptors between brain and periphery is an outdated and incorrect concept.

To the best of my knowledge, Proglumide is a non-selective cholecystokinin (CCK) antagonist. However Authors refer to this drug as a potent CCKAR, which is not true.

The increase of free fatty acids (FFA) in the bronchoalveolar lavage (BAL) fluid of HFD-fed obese is not impressive. How that compares with the concentration of FFA applied to in vitro experiments? Also, the rationale for measuring only in BAL is unclear.

The effect of CCKR antagonists in vivo is rather small and one would not expect that to reach statistical significance. I could not find statistical details, other than $*p < X$. This must be fixed.

It would have been interesting to see the effect of a CCK agonist in lean mice. Do Authors expect that to have an effect?

The rationale for administering drug intranasally is not clear. Readers would benefit on some more explanation on rationale here. Along these lines, how confident are the authors that their chosen route of administration delivers the drug as they intended to? The dosage of proglumide (50x3) is rather high and proglumide is orally active. Furthermore, can the authors exclude any central effect when delivering drugs intranasally?

While it is interesting that IL17a do not change, this is far too little to rule out modulation of inflammation, which is clearly an important factor. Evidence exists that CCK signalling can modulate inflammation and it is unclear why Authors have not included a multiplex experiment or screening of additional inflammatory markers on lungs. These are not difficult experiments.

The image showing the isotype control is difficult to interpret. Higher magnification images are needed.

Most images lack scale bar.

Reviewer #4 (Remarks to the Author):

The study demonstrated the presence of CCK and CCKAR in ASM of mice and humans. In mice and cultured cells CCK was capable of inducing ASM contraction. In obese mice, there was increased expression of CCK and ASM hyperresponsiveness. The inhibition of CCKAR using pharmacological and genetic approaches reduced ASM hypercontractility. Authors propose that CCKAR antagonists could be useful to treat patients with asthma.

This is an interesting study that provides preliminary evidence for a function of CCK that has been ignored in the past. However, there are several weaknesses and inconsistencies that should be dealt with.

Major

1. In the introduction and in the first set of results, authors give the impression that only one study has evaluated CCK in the lung (ref 37, Stretton). This is not true. There are several experimental and clinical studies that have evaluated the presence of components of the CCK pathway in the lung. In humans, for example, there are radiotracers that detect CCK receptors and

are used to detect lung inflammation and neoplasia. In animals, several studies have been published. Authors should provide proper reference to all these findings.

2. There seem to be some controversy regarding the site of expression of CCK receptors in the lungs. Cong et al. (PMID12800239) reported the expression of CCKAR and CCKBR in the lung predominantly in vascular endothelial cells, macrophages, bronchial epithelial cell and alveolar cells. Xu et al, (PMID15456538) reported the presence of CCKAR and CCKBR in lung macrophages. Li et al. also report CCK receptors in lung macrophages. Authors should reconcile their finding with the existing published data.

3. Single cell RNA seq provide state of the art method for defining transcript expression in distinct cell types. Authors should consider using this method to explore the expression of CCK and CCK receptors in the mouse and human lungs. The RNA seq that is cited as the basis for the identification of CCK receptors in the lung was prepared from ASM and cannot be regarded as representative of a lung (Himes et al, Ref 41).

4. The claim that CCK is expressed in the lung is fragile. It was obtained from cell cultures and only the transcript was determined. Authors should provide undisputed evidence for the presence of CCK peptide in the lung. Moreover, functional studies should be performed to explore physiological mechanisms that control lung CCK.

5. CCK is mostly produced in the gut in response to food intake. In this study authors claim that the lungs have an autonomous CCK system controlling ASM. There are no experiments that clearly isolate the gut from the putative lung CCK system.

6. Another important weakness of the study is that CCK acts in the brain to control autonomous responses. It has been shown that metabolic hormones can control the airway reactivity by signaling in brain stem cholinergic neurons. In order to support their claim that the lung has a CCK autonomous system they should provide evidence that it works separately from any CCK action in the brain.

Minor

1. Patients with asthma, should not be referred as asthmatics.
2. Abbreviations in the figures should be defined in the legends.
3. Authors should use dot plot plus SEM in all graphs (instead of bars).

We thank the reviewers for their helpful and constructive comments. Changes made to the manuscript are highlighted (either with vertical line to the left of revised paragraphs, or boxed for text). Below I detail our responses to each of the critiques. For clarity, reviewer comments are italicized and followed by our specific responses.

Response to comments by reviewer #1

This is a 4 Figure manuscript submitted to Nature Communications investigating the role of CCK in obesity induced asthma. Authors use a combination of methods including genetic ablation and pharmacological intervention to show that the CCK/CCKAR axis contracts ASM cells in an autocrine manner, in a FFA-dependent mechanism. Authors further found that CCK is elevated in the lungs of mouse models of obesity-associated asthma, mediating AHR. The paper reads well. My major concern is the lack of immunology in the study, since authors conclude that CCK contracts ASM independently of the lung microenvironment (including immune cells). Please find my comments below.

Response: Thank you. We have now addressed the lack of immunology with new experiments – see details below.

Major comments.

1.) The big picture of the manuscript is the fact that CCK contracts ASM and induces AHR. This is why only 24 hours of treatment (Figure 4) is required to abrogate AHR using antagonists. However, obesity-induced asthma is mainly driven by the immune system. Authors try to look at TH17 but only say that there is no difference and the data is not shown. This reviewer believes that authors should study the immune system of these mice. What are the effects of the treatments on the immune system? Have the authors looked at inflammatory cells in the BAL? Hypothetically, CCK acts independently of the immune system, but authors should show it. This topic is briefly discussed in the discussion, but authors should show data in order to draw this conclusion.

Response: To address this concern, we first measured the total BAL cell counts in the 4 groups of mice (chow-vehicle; chow-devazepide; HFD-vehicle; and HFD-devazepide). We found no significant changes in the total BAL cell count in either HFD-fed obese mice compared to regular-chow-fed mice (which is not surprising as these were naïve mice - not challenged with allergens), or by the antagonist treatment. We also performed BAL cell differential counts and found that CCK antagonist treatment did not cause any changes in the percentages of macrophages, lymphocytes, eosinophils, or neutrophils. We also performed BAL cell differential counts on WT-vehicle, WT-proglumide, *db/db*-vehicle, and *db/db*-proglumide mice. Although we found that eosinophils are increased in the BAL of *db/db* mice compared to WT mice, proglumide did not significantly alter the level of eosinophils in *db/db* mice compared to saline treatment group. These data are now included in **Supplementary Fig. S3**.

Using a multiplex assay, we also assessed the levels of a panel of inflammatory molecules/cytokines in the BAL from the 4 groups of mice (chow-vehicle; chow-devazepide; HFD-vehicle; and HFD-devazepide.). We found that cytokines in BAL were not altered in obese mice compared to their controls. We also found that devazepide did not alter the levels of

cytokines in HFD-fed mice. Similarly, we also performed BAL cell differential counts on WT-vehicle, WT-proglumide, *db/db*-vehicle, and *db/db*-proglumide mice. Among the cytokines increased in BAL of *db/db* mice compared to saline control include eotaxin, which is a specific chemoattractant for eosinophil cells to sites of inflammation (PMID: 9379061; PMID: 7509365; PMID: 8597956). However, similar to the absence of significant change in the level of eosinophils in proglumide-treated *db/db* mice, there was also no change in the eotaxin level in proglumide-treated *db/db* mice compared to saline-treated *db/db* mice. We also observed that *db/db* mice have higher level of MIP-1a in BAL compared to wild type mice, which was again not altered by proglumide treatment. These data are now included in **Supplementary Fig. S4**. Together these data support that antagonizing CCKAR abolishes the innate airway hyperresponsiveness in obese mice without affecting airway inflammation.

2.) *Although previously published, it would help the reader if authors show some of the RNAseq data in Figure 1 which led to focus on CCK.*

Response: We have now listed the top GPCR genes (Gene ID and FPKM values) from the RNA Seq dataset in **Fig. 1a**.

3.) *The text describing Figure 1 is lacking details. It would help the reader if authors describe the experiment that was performed. For example, what is d0, d14, HBE in Figure 1a? The order of graphs is misleading in Figure 1 and 2.*

Response: We have re-written the text describing results in Fig.1. We also included the details of the abbreviations in Fig 1. HBE (human bronchial epithelial) cells were cultured at air-liquid interface which allowed their differentiation into pseudostratified columnar epithelium that recapitulates *in vivo* airway epithelium. D0 and D14 refer to the days of HBE differentiation. HBE_D0 cells are not differentiated, D14 cells are fully differentiated. We have rearranged the panels in Figs. 1 and 2 so they are easier to follow now.

4.) *In Figure 1c, Acetylcholine is used as a positive control. Would A71623 enhance the stiffness caused by Acetylcholine?*

Response: We conducted additional contractility experiment with Traction Force Microscopy to test whether A71623 enhances the effect of Acetylcholine. As we have shown previously, A71623 agent, induces contraction of ASM cells. When given together, A71623 and Ach, led to greater ASM contraction than either compound alone. This data is now included in **Fig. 1 e,f**. We note that due to the pending retirement of co-author Dr. Jeffrey Fredberg whose lab provided the original contractility data this new experiment was performed by Nicole Schaible in Dr. Rama Krishnan's lab at Beth Israel Deacones Medical Center of Harvard Medical School. Nicole Schaible and Rama Krishnan are now included as new co-authors.

5.) *Related to comment 3.). All of the CCK data is based on RNAseq or qPCR except Figure 3d which is an ELISA. What is the rationale for combining these readouts? Authors should comment.*

Response: We assessed CCK differential gene expression mainly by measuring changes in the mRNA transcript by qPCR or by re-evaluating RNA-Seq data. In Fig. 3d, we wanted to provide additional evidence that free fatty acids (palmitoleic and myristolic acids) upregulate CCK by measuring expression at the protein level. Moreover, we wanted to show that CCK, which is a secreted hormone, was actually secreted by ASM cells following free fatty acid treatment, hence ELISA was performed in cell culture medium. We now also provide new data on CCK protein level (in addition to CCK mRNA data) in mouse lungs (**Fig.3f,h**).

6.) *Authors discuss the role of circulating CCK in the discussion. However they do not put it in relation to their findings. What is the role of circulating CCK in their model? Is it high? Is the autocrine role of CCK in ASM more important? Are circulating CCK and FFA both involved? Authors should clarify.*

Response: Our current *in vivo* data do not distinguish circulating CCK from autocrine CCK; thus we could not infer which CCK source is more important in ASM contraction. This could be delineated using ASM-specific CCK knockout obese mice which to the best of our knowledge do not exist. We envision that both circulating and ASM-secreted CCK play roles in the contraction of ASM. We also think that circulating FFA, like circulating CCK, is involved in CCKAR-mediated ASM contraction; in the case of FFA, this would be via upregulating lung CCK which can then engage the CCKAR receptor in ASM. We have included these points in the discussion (lines 266-278).

7.) *In Figure 4, some controls are missing. Chow/vehicle should be in all plots to show the basal levels of lung resistance. In addition, how did mice react to 100 mg/mL methacholine in Figure 4e? This is important since all other plots show up to 100 mg/mL methacholine.*

Response: We have now included 1) the regular chow-fed mice (with or without CCK antagonist treatment) as controls for the HFD-fed mice (**Fig.4b,c**), and 2) wild type (WT) mice (with or without CCK antagonist treatment) for the *db/db* mice (**Fig.4e,f**). These new controls provided baseline lung resistance comparisons. Importantly CCK antagonist treatment did not alter the baseline lung resistance (RL) in these controls where CCK level was not elevated. In Fig. 4e, the effect of proglumide in AHR was significant in *db/db* mice versus wild type controls at 25 mg/ml but not at 100 mg/ml methacholine, likely due to an outlier mouse in the treatment group at the high methacholine concentration.

Minor comments.

Line 31. Twice “for obsess asthmatics”

Response: We have now corrected the error.

It is surprising to see a graphical summary in Figure 4 of a Nature communications manuscript

Response: We included the graphical summary to help readers better understand our findings. If the Journal does not allow it we will remove it.

The ticks and methacholine doses are not in line in Figure 4

Response: We have now corrected these errors.

What is “Base” versus “0” in Figure 4?

Response: Baseline refers to the measurement of total lung resistance in the absence of nebulization. “0” refers to the nebulization with PBS, in which methacholine was dissolved.

Fig 3b missing scale bar

Response: We have now included scale bars in the images.

Response to comments by reviewer #2

In this manuscript the authors demonstrate a novel therapeutic target for treating obese asthmatics via antagonizing cholecystokinin (CCK) and its receptor CCKAR signaling in the airway smooth muscle cells. They first verified that CCK and CCKAR are highly expressed in airway smooth muscle cells but not in other types of airway residential cells such as fibroblasts and epithelial cells. Then, they demonstrate that, in both in vitro cell culture and in vivo mouse models, induction of CCK and activation of CCKAR led to increased contractility of airway smooth muscle cells in vitro and airway hyperresponsiveness (AHR) in vivo, which are all reversed by antagonizing CCKAR with known drug agents. These data seem to be very convincing that CCK/CCKAR antagonists may bring a great hope to those suffering from obesity-associated asthma because that this subtype of asthma is notorious for poor response to currently available therapy.

Overall, this is a well thought out and carefully designed study. The experimental coverage is quite thorough and the results appear to be technically sound with well-recognized techniques. Particularly in regarding the assays of ASMC stiffness and contraction using optical magnetic twisting cytometry (OMTC) and traction force microscopy (TFM), these results are very robust and trustworthy, which is not surprising because these two techniques are actually originated from one of the author’s affiliated labs. Moreover, these two techniques have now been widely used in many research groups throughout the world and in many studies of airway smooth muscle cell mechanics in relation to diseases including but not limited to asthma. In this reviewer’s eyes, the data of ASMC response in stiffness and contraction force to CCK/CCKAR stimulation are consistent and comparable to previously reported ASMC responses to other contractile agonists in the literature.

Response: Thank you.

There are only a couple of minor issues related to the OMTC protocol used in this study that may need the authors to clarify. 1) In this study, the beads were coated with non-specific binding poly-L-lysine instead of the more commonly used, specific integrin binding RGD peptide. What is

the reason to change the surface coating agent for the magnetic beads? 2) the twisting frequency of magnetic field was set at 0.77 Hz, instead of usually used 0.3-0.5 Hz to mimic normal breathing rate. Any particular consideration of this setting?

Response: We provide the following clarifications.

1. We used poly-L-lysine (PLL) instead of RGD peptide in order to measure cell mechanics as cleanly as possible. Zhou et al (2009, PMID: 19520830) used PLL coating in OMTC assay to investigate the physical nature of the cytoplasm under compression. PLL coating binds beads on cell surface through electrostatic force, in contrast to RGD coating that connects beads with intracellular cytoskeleton through integrin (or focal adhesion contacts) which can affect integrin signaling. Because it was unknown whether addition of CCK agonist A71623 would affect integrin or focal adhesion signaling, especially in the context of cell stiffness, we used PLL coated beads to measure the mechanical effects of A71623 treatment.
2. Although the twisting frequency happens to be close to breathing frequency, it was not intended to mimic breathing. If we measured the mechanics in the context of breathing, then we would have controlled the frequency as well as the magnitude of deformation. During tidal breathing or occasional deep breathing, the airways and the cells within airways undergo 4~10% stretch every 2-3 seconds. Here, we want to measure the change in cell stiffness in response to CCK agonist by twisting beads which would cause local deformation around beads in the order of 1 um or less. Such twisting frequency of 0.77 Hz has been used in other studies with similar objectives (Zhou et al 2012, PMID: 22171066; Zhou et al 2009, PMID: 19520830; Vahabikashi et al 2019 PMID: 30685055).

The writing of the manuscript is in general of high quality with clarity and professionalism, except for a couple of minor typos such as 1) at the end of line 123, the “of” is redundant, 2) on line 294, “ALI” needs full spelling since it first appeared here, 3) on line 457, it should include “LORD: lorglumide”, 4) once abbreviation of “ASM” was defined, it is better to replace all occurrences of “airway smooth muscle” afterwards throughout the manuscript.

Response: We have now corrected these errors/typos as suggested.

Response to comments by reviewer #3

Panganiban and colleagues report a tidy and logical series of experiments that demonstrate that cholecystokinin (CCK) and its receptor CCKR1 is increased in the lungs of obese mice and that pharmacological blockade of CCK signalling reduce obesity-associated airway hyper-responsiveness (AHR). More specifically, authors suggest that CCK and CCKR1 are expressed by smooth muscle cells and regulate their contractility via a autocrine mechanism. The manuscript is interesting and overall easy to read.

Response: Thank you.

Below are some specific points that Authors might consider addressing and/or clarifying. The abstract contains a few typos and repetitions.

Response: We have corrected these typos/errors in the Abstract and in other sections.

Strictly speaking CCK it is known as a satiety signal, with minimal, if any, effect on satiety. Authors should consider a more rigorous description of CCK and its actions. Also, it is suggested that Authors leverage on more recent references. For example, the statement "In the CNS, CCK acts as a neurotransmitter to regulate satiety and nociception" is not only limited, but also supported by a review published 40 years ago. Likewise, the segregation of CCK1 and CCK2 receptors between brain and periphery is an outdated and incorrect concept.

Response: We have revised our description of CCK actions and expanded our introduction by including references pertaining to recent relevant findings on CCK and CCKAR in the lung (Lines 67-84).

To the best of my knowledge, Proglumide is a non-selective cholecystikinin (CCK) antagonist. However, Authors refer to this drug as a potent CCKAR, which is not true.

Response: We have corrected this error. Compared to proglumide which is a pan-CCK antagonist, devazepide is much more specific and potent for CCKAR. We have made this clear in the revised text.

The increase of free fatty acids (FFA) in the bronchioalveolar lavage (BAL) fluid of HFD-fed obese mice is not impressive. How does that compare with the concentration of FFA applied to in vitro experiments? Also, the rationale for measuring only in BAL is unclear.

Response: We have obtained additional new data that directly measured FFAs in both the serum and the lungs of obese and control mice (**Supplementary Fig.S2**). The data show increased levels of FFAs in obese mice as compared to that in control non-obese mice. Such increases correlate with elevated CCK levels in the lungs of obese mice (both HFD-fed and *db/db* mice versus their controls). The FFA concentration in the lungs is about 1 μmol per gram of lung tissue. FFA concentration in serum is about a few hundred μM , which is comparable to that (250 μM) used in the in vitro ASM experiment.

*The effect of CCKR antagonists in vivo is rather small and one would not expect that to reach statistical significance. I could not find statistical details, other than $*p < X$. This must be fixed.*

Response: The difference in pulmonary resistance (R_L) between obese mice and their lean controls is not big in general. For example, the R_L is about 3.5-4 $\text{cm H}_2\text{O}/\text{ml}/\text{s}$ for HFD mice and around 3 $\text{cm H}_2\text{O}/\text{ml}/\text{s}$ for regular chow-fed mice at the highest methacholine concentration. The difference is even smaller at lower concentrations of methacholine. Such small window, together with the individual variability in mice, makes the experiments very challenging. Nevertheless, in all experiments we observed reduced R_L in obese mice treated with CCK antagonists. The p values were obtained using two-way ANOVA. The method is included in the Methods section.

It would have been interesting to see the effect of a CCK agonist in lean mice. Do Authors expect that to have an effect?

Response: Our current results show that CCK can induce ASM contraction and contributes to obesity-associated innate airway hyperresponsiveness. Stretton and Barnes also showed that a CCK octapeptide constricts both guinea pig and human airway tissue slices (PMID: 2758237). We anticipate that administering CCK agonist will increase airway contraction in lean mice *in vivo*.

The rationale for administering drug intranasally is not clear. Readers would benefit on some more explanation on rationale here. Along these lines, how confident are the authors that their chosen route of administration deliver the drug as they intended to? The dosage of proglumide (50x3) is rather high and proglumide is orally active. Furthermore, can the authors exclude any central effect when delivering drugs intranasally?

Response: The main reason for using intranasal drug administration as the route of delivery of CCKAR antagonists is that many asthma drugs are administered as nasal sprays. We also reasoned that this route would be the most efficient and direct way in delivering the antagonists to the intended target (airway smooth muscle). Intratracheal administration could also deliver to the airways, but it is rather invasive and has the potential to increase airway reactivity. Another reason for intranasal administration of CCKAR antagonists is to minimize the possible undesired effects of inhibiting CCK/CCKAR systemically in the obese. We did try delivering CCKAR antagonists via intraperitoneally injection which did not result in statistically significant reduction of innate airway hyperresponsiveness (data not included in the manuscript), possibly due to the reasons stated above. While we aim to deliver CCKAR antagonists to the airway smooth muscle as efficiently as possible to antagonize the innate airway hyperresponsiveness in the obese mice, we cannot completely exclude the possibility that intranasal administration has effect on non-pulmonary tissues.

While it is interesting that IL17a do not change, this is far too little to rule out modulation of inflammation, which is clearly an important factor. Evidence exist that CCK signalling can modulate inflammation and it is unclear why Authors have not included a multiplex experiment or screening of additional inflammatory markers on lungs. These are not difficult experiments.

Response: We have addressed a similar critique from Reviewer #1. Here is the copy of the response:

To address this concern, we first measured the total BAL cell counts in the 4 groups of mice (chow-vehicle; chow-devazepide; HFD-vehicle; and HFD-devazepide). We found no significant changes in the total BAL cell count in either HFD-fed obese mice compared to regular-chow-fed mice (which is not surprising as these were naïve mice - not challenged with allergens), or by the antagonist treatment. We also performed BAL cell differential counts and found that CCK antagonist treatment did not cause any changes in the percentages of macrophages, lymphocytes, eosinophils, or neutrophils. We also performed BAL cell differential counts on WT-vehicle, WT-proglumide, *db/db*-vehicle, and *db/db*-proglumide mice. Although we found that eosinophils are increased in the BAL of *db/db* mice compared to WT mice, proglumide did not significantly alter the level of eosinophils in *db/db* mice compared to saline treatment group. These data are now included in **Supplementary Fig. S3**.

Using a multiplex assay, we also assessed the levels of a panel of inflammatory molecules/cytokines in the BAL from the 4 groups of mice (chow-vehicle; chow-devazepide;

HFD-vehicle; and HFD-devazepide.). We found that cytokines in BAL were not altered in obese mice compared to their controls. We also found that devazepide did not alter the levels of cytokines in HFD-fed mice. Similarly, we also performed BAL cell differential counts on WT-vehicle, WT-proglumide, *db/db*-vehicle, and *db/db*-proglumide mice. Among the cytokines increased in BAL of *db/db* mice compared to saline control include eotaxin, which is a specific chemoattractant for eosinophil cells to sites of inflammation (PMID: 9379061; PMID: 7509365; PMID: 8597956). However, similar to the absence of significant change in the level of eosinophils in proglumide-treated *db/db* mice, there was also no change in the eotaxin level in proglumide-treated *db/db* mice compared to saline-treated *db/db* mice. We also observed that *db/db* mice have higher level of MIP-1a in BAL compared to wild type mice, which was again not altered by proglumide treatment. These data are now included in **Supplementary Fig. S4**. Together these data support that antagonizing CCKAR abolishes the innate airway hyperresponsiveness in obese mice without affecting airway inflammation.

The image showing the isotype control is difficult to interpret.

Response: We have now included a higher magnification (63x, oil) of image for the isotype control. Please see Supplementary **Fig. S1**.

Most images lack scale bar.

Response: We have now included scale bars for all images.

Response to comments by reviewer #4

The study demonstrated the presence of CCK and CCKAR in ASM of mice and humans. In mice and cultured cells CCK was capable of inducing ASM contraction. In obese mice, there was increased expression of CCK and ASM hyperresponsiveness. The inhibition of CCKAR using pharmacological and genetic approaches reduced ASM hypercontractility. Authors propose that CCKAR antagonists could be useful to treat patients with asthma.

This is an interesting study that provides preliminary evidence for a function of CCK that has been ignored in the past. However, there are several weaknesses and inconsistencies that should be dealt with.

Response: Thank you. Please see our responses below that address the weaknesses and inconsistencies.

Major

1. In the introduction and in the first set of results, authors give the impression that only one study has evaluated CCK in the lung (ref 37, Stretton). This is not true. There are several experimental and clinical studies that have evaluated the presence of components of the CCK pathway in the lung. In humans, for example, there are radiotracers that detect CCK receptors and are used to detect lung inflammation and neoplasia. In animals, several studies have been published. Authors should provide proper reference to all these findings.

Response: We have revised our description of CCK functions and expanded our introduction by including references pertaining to recent relevant findings on CCK and CCKAR in the lung (Lines 67-84).

2. There seem to be some controversy regarding the site of expression of CCK receptors in the lungs. Cong et al. (PMID12800239) reported the expression of CCKAR and CCKBR in the lung predominantly in vascular endothelial cells, macrophages, bronchial epithelial cell and alveolar cells. Xu et al, (PMID15456538) reported the presence of CCKAR and CCKBR in lung macrophages. Li et al. also report CCK receptors in lung macrophages. Authors should reconcile their finding with the existing published data.

Response: We have cited these published studies in the introduction (lines 77-80). We also added a new paragraph in the discussion section (lines 264-278) on the implications of these studies in the context of our findings in ASM cells.

3. Single cell RNA seq provide state of the art method for defining transcript expression in distinct cell types. Authors should consider using this method to explore the expression of CCK and CCK receptors in the mouse and human lungs. The RNA seq that is cited as the basis for the identification of CCK receptors in the lung was prepared from ASM and cannot be regarded as representative of a lung (Himes et al, Ref 41).

Response: As stated in our manuscript, we re-evaluated an already existing RNA-Seq dataset in ASM cells, the main effector of bronchoconstriction, for expression of GPCRs, which are the direct targets of the majority of asthma drugs. While we agree that single cell RNA-seq provides state-of-the-art method for defining transcript expression in distinct cell types in the lung, we believe such comprehensive omics studies in the whole lung are beyond the scope of our current ASM-focused study.

4. The claim that CCK is expressed in the lung is fragile. It was obtained from cell cultures and only the transcript was determined. Authors should provide undisputed evidence for the presence of CCK peptide in the lung. Moreover, functional studies should be performed to explore physiological mechanisms that control lung CCK.

Response: We now provide additional new data (**Fig. 3f,h**) showing the presence of CCK peptide in the lungs of obese mice (HFD and *db/db*) as measured by ELISA. Consistent with our finding, the Human Protein Atlas database showed CCK protein expression in the bronchus (<https://www.proteinatlas.org/ENSG00000187094-CCK/tissue>). CCK expression is likely regulated physiologically by free fatty acids (FFA) as we showed 1) some long chain FFA upregulated CCK expression (both mRNA and protein) in ASM cells (**Fig.3c,d**), and 2) there is a concomitant FFA increase in obese mice (**Supplementary Fig.S2**).

5. CCK is mostly produced in the gut in response to food intake. In this study authors claim that the lungs have an autonomous CCK system controlling ASM. There are no experiments that clearly isolate the gut from the putative lung CCK system.

Response: Although we showed that CCK and CCKAR are expressed in the lung, especially in ASM cells, we do not discount the possibility that CCK in the lung can come from the gut or from other extrapulmonary tissues that have reached the airways. The circulating CCK could also contribute airway hyperresponsiveness in obese mice as shown by our schematic diagram in Fig. 4g.

6. Another important weakness of the study is that CCK acts in the brain to control autonomous responses. It has been shown that metabolic hormones can control the airway reactivity by signaling in brain stem cholinergic neurons. In order to support their claim that the lung has a CCK autonomous system they should provide evidence that it works separately from any CCK action in the brain.

Response: We agree that CCK acting in the brain can potentially control airway reactivity through signaling in the cholinergic neurons of brain stem. In the same manner as discussed above (Comment #5), circulating CCK likely contributes to obesity-associated AHR. Our work showed that CCK and CCKAR are also expressed in ASM cells and that, as shown by biomechanical (OMTC and TFM experiments) and *in vivo* experiments, this CCK/CCKAR signaling is functionally important for ASM contractility and obesity-induced AHR. Our study, however, does not attempt to prove that obesity-associated AHR is solely mediated by elevated CCK/CCKAR signaling in the ASM. It is likely that all or a combination of these CCK sources contribute to the complex, obesity-associated AHR. Future studies aimed to delineate the contributions of CCK from different sources or tissues may be facilitated by the generation of ASM- or lung-specific CCK/CCKAR knockout mice.

Minor

1. Patients with asthma, should not be referred as asthmatics.

Response: We have corrected this throughout the manuscript.

2. Abbreviations in the figures should be defined in the legends.

Response: We have now defined the abbreviations in the figure legends.

3. Authors should use dot plot plus SEM in all graphs (instead of bars).

Response: We have done this for all graphs as suggested.

REVIEWER COMMENTS

Reviewer #1 (Remarks to the Author):

The authors addressed all the concerns and the revised manuscript is now much stronger.

Reviewer #2 (Remarks to the Author):

The responses to the reviewer's comments and corresponding revision of the manuscript are satisfactory. In this reviewer's opinion, this manuscript is now suitable for publication in Nat Comm.

Reviewer #3 (Remarks to the Author):

The authors have addressed almost all my points. The MS has improved substantially.

My comment on CCK and CCKR function and distribution was overall addressed, however, the authors still refer to CCK as a satiety factor. As mentioned, CCK signal satiation. I appreciate that the auto-correction has changed satiation with station on my original comment and I apologize for that.

The full description of statistic is still missing. It is advisable that F values for ANOVA are reported for individual variables and their interaction.

Reviewer #4 (Remarks to the Author):

Authors provided satisfactory answers to only two out of six major problems pointed during the first round of review.

Question #1
OK

Question #2
I have asked them to reconcile their data with the apparently conflicting data regarding CCK and CCK receptors published previously. Authors have only included new references but no in depth discussion of the issue was provided.

Question #3
Not dealt with

Question #4
Provided new data, OK

Question #5
Not dealt with

Question #6
Not dealt with

We once again thank the reviewers for their comments during the second review. Changes made to the manuscript are highlighted (**boxed** for text). Below I detail our responses to each of the remaining critiques. For clarity, reviewer comments are italicized and followed by our specific responses.

Response to comments by Reviewer #1

The authors addressed all the concerns and the revised manuscript is now much stronger.

Response: Thank you.

Response to comments by Reviewer #2

The responses to the reviewer's comments and corresponding revision of the manuscript are satisfactory. In this reviewer's opinion, this manuscript is now suitable for publication in Nat Comm.

Response: Thank you.

Response to comments by Reviewer #3

1. The authors have addressed almost all my points. The MS has improved substantially.

Response: Thank you.

2. My comment on CCK and CCKR function and distribution was overall addressed, however, the authors still refer to CCK as a satiety factor. As mentioned, CCK signal satiation. I appreciate that the auto-correction has changed satiation with station on my original comment and I apologize for that.

Response: We have now refrained from referring CCK as a satiety factor. As reflected in lines 33 and 70-72, we now refer to CCK as a hormone that signals satiation.

3. The full description of statistic is still missing. It is advisable that F values for ANOVA are reported for individual variables and their interaction.

Response: We have now modified the description of the statistics (line 517). F values were reported for treatment, methacholine, and their interaction for animal studies in Figure 4 legend (lines 577-597).

Response to comments by Reviewer #4

Authors provided satisfactory answers to only two out of six major problems pointed during the first round of review.

Response: Please see below additional responses to each of the remaining questions.

Question #1

OK

Response: Thank you.

Question #2

I have asked them to reconcile their data with the apparently conflicting data regarding CCK and CCK receptors published previously. Authors have only included new references but no in depth discussion of the issue was provided.

Response: We have now added a new paragraph in the discussion section (lines 274-294) that reconcile our finding on the expression and function of CCK and CCKAR in ASM cells with existing published reports on the expression and possible function of CCK and CCK receptors in other lung cell types. In summary, our study focuses on ASM cells but it does not contradict the expression of CCK and CCKAR in other lung cell types as reported by previous studies given that our experiments were not designed to demonstrate the expression of CCK and CCKAR solely in the ASM cells. However, these previous studies were conducted in the context of other lung diseases (e.g. endotoxin shock and lung cancer) with findings that could not readily be applied in the context of obesity-associated asthma, the disease focus of our research. Thus, while CCK receptors are also expressed in non-ASM cells in which they mediate cellular processes in the context of other lung disorders, CCKAR signaling in ASM cells appears to be the major driver of obesity-associated AHR.

Question #3

Not dealt with

(original comment): Single cell RNA seq provide state of the art method for defining transcript expression in distinct cell types. Authors should consider using this method to explore the expression of CCK and CCK receptors in the mouse and human lungs. The RNA seq that is cited as the basis for the identification of CCK receptors in the lung was prepared from ASM and cannot be regarded as representative of a lung (Himes et al, Ref 41).

Response: We did explore existing multiple single cell RNA seq databases, including the Single Cell Portal (singlecell.broadinstitute.org) and the Human Protein Atlas (which recently added both protein and RNAseq data on single cell types). However, the data on ASM cells are sparse in these existing single cell datasets and are not of sufficient resolution to allow us to compare CCKAR/CCK expression among different lung cell types. At present time our RNA-seq dataset plus qRT-PCR-based validation provided high-confidence data on the expression of the single genes (CCK/CCKAR). With the cost of single cell RNA-seq coming down and coverage of relatively low expression genes increasing, it is possible in the future to more comprehensively define the transcript expression in distinct cell types (including ASM) in the lung; however, we

do believe such extensive omics studies in the whole lung or trachea are beyond the scope of our current ASM-focused study.

Question #4

Provided new data, OK

Response: Thank you.

Question #5

Not dealt with

(original comment): CCK is mostly produced in the gut in response to food intake. In this study authors claim that the lungs have an autonomous CCK system controlling ASM. There are no experiments that clearly isolate the gut from the putative lung CCK system.

Response: Although we showed that CCK and CCKAR are expressed in the lung, especially in ASM cells, we do not discount the possibility that CCK in the lung can come from the gut-derived CCK that has reached the airways. We consider this as a limitation of our study; a discussion of which has now been included in lines 295-313. Our experimental results indicate that CCK is produced in the lungs as shown by our qRT-PCR, immunostaining, and ELISA data. Our results, however, do not identify which CCK source (circulating or ASM-secreted) contributes more to obesity-induced AHR, a hallmark of asthma. It is likely that both CCK sources, independently and/or synergistically, lead to the ASM contraction once the receptors (i.e. CCKAR) on the surface of these smooth muscle cells are engaged. Similarly, circulating FFAs may initiate ASM contraction when they induce the production and the secretion of CCK (again, either in the gut or in the lung) that engages CCKAR in ASM. However, regardless of the CCK source, the resulting ASM contraction and AHR may be alleviated by blocking CCK/CCKAR signaling with the antagonists.

Question #6

Not dealt with

(original comment): Another important weakness of the study is that CCK acts in the brain to control autonomous responses. It has been shown that metabolic hormones can control the airway reactivity by signaling in brain stem cholinergic neurons. In order to support their claim that the lung has a CCK autonomous system they should provide evidence that it works separately from any CCK action in the brain...

Response: It has been shown that metabolic hormones (e.g. leptin and insulin) can regulate AHR in obese mice by acting on cholinergic neurons in the brainstem, though this activity has not been reported for CCK. Although CCK levels are elevated in obese rodents or after high-fat diet intake, CCK itself is not able to cross the blood-brain barrier unlike leptin and insulin. Its potential action in the brainstem-mediated control of obesity-associated AHR, if at all, would likely have to be through brainstem-derived CCK. However, previously published reports of the level of CCK in the brain of obese mice (*ob/ob*) showed either no significant difference or lower compared to lean controls (Refs. 69-71: Hanksy J, et.al., Aust J Exp Biol Med Sci; Straus et.al., 1979, Science; Lavine et. al., 2010 Endocrinology). Furthermore, another published study (Ref. 72: Saito et al.1981, Endocrinology) showed that the level of binding of CCK to its receptors in

the brain of *db/db* mice is unchanged, suggesting that there is no increased CCKAR signaling in the brain that could have contributed to the increased AHR of the *db/db* mice that we observed in our study. In order to completely rule out the possible role of CCK/CCKAR in the brainstem in modulating obesity-associated AHR, conditional tissue-specific CCK or CCKAR knockout in obese mice is needed; however, this is beyond the immediate scope of our present study. We consider this a potential limitation of our study and provided a detailed discussion in lines 314-327 to reflect that.

REVIEWERS' COMMENTS

Reviewer #4 (Remarks to the Author):

I have no further questions.